# ReCogLab: a framework testing relational reasoning & cognitive hypotheses on LLMs

Andrew Liu[1], Henry Prior[1], Gargi Balasubramaniam[1], Rivka Moroshko[1], Amir Zait[1], Ilia Labzovsky[1], Danny Karmon[1], Ishita Dasgupta[1], Kim Stachenfeld[1], and Kenneth Marino[1]

[1]Google Deepmind, {ahliu, hprior, gargisb, rikimoroshko, amirzait, ilabz, dannykarmon, idg, stachenfeld, kmarino}@google.com

## Abstract

A fundamental part of human cognition is the ability to not only recall previous memories, but also reason across them to draw conclusions. In cognitive science and psychology, this is termed relational reasoning and a number of effects and biases have been observed in human cognition. Designing experiments to measure these reasoning effects is effortful, and does not transfer easily to analyzing language model reasoning patterns. To make exploring language models on relational reasoning easier, we introduce ReCogLab – a generative framework for constructing reasoning examples. Unlike static datasets, our framework has a number of benefits that help us in our goal of flexible evaluation of LLMs. First, our framework allows us to control the difficulty and context-length of the problem, allowing us to scale with model capability and evaluate LLMs at a variety of scales. Second, the ability to change the configuration of a dataset dynamically allows us to probe models on different aspects and capabilities. Finally, the flexibility of our approach enables the recreation of classic cognitive science experiments and the systematic study of relational reasoning biases in language models. We demonstrate several such experiments and present our findings on a wide variety of open and closed-source language models. We release all data and code at https://github.com/google-deepmind/recoglab.

## 1 Introduction

While recent work on memory in large language models (LLMs) assumes that memory refers to recall of a specific piece of information from the past (Li et al., 2024a; Levy et al., 2024), research into human memory and reasoning has long shown memory to be much more complex. When humans remember events, facts, places, they don't just recall disconnected pieces, they recall associations (Cohen, 1993; Eichenbaum, 2004). Thus humans have the remarkable ability to recall relationships ("relational memory") and draw inferences across related memories ("relational reasoning").

Motivated by the reasoning and memory abilities of language models, prior works have created new reasoning benchmarks by defining new problems and collecting data targeting specific purposes like visual reasoning (Johnson et al., 2017; Kahou et al., 2018; Wang et al., 2024), math reasoning (Lewkowycz et al., 2022; Wang et al., 2024), rule-following (Kazemi et al., 2023; Clark et al., 2020b), or story problems (Yang et al., 2024; Yu et al., 2020) – resulting in a fixed test dataset. These datasets are subsequently benchmarked against current state-of-the-art models to gauge capability progression on the specific problem. However, this approach has numerous problems. First, these datasets, like all static datasets are **susceptible to increasing model performance**. Even as dataset creators try to invent harder problems that are quite difficult for current models, the rapid progress of the field often overtakes the dataset, giving it a short shelf-life for useful benchmarking. Second, depending on the design of the benchmark, **it can often be quite difficult to probe** or tease apart what makes the problem difficult for models or to explore different capabilities of the model. For example, if reasoning requires both recalling information and logical deduction, can we easily separate whether a model fails due to memory recall or poor reasoning? Finally, these datasets are **fixed and cannot be used by later authors to study related phenomenon**. In the Cognitive Sciences for instance, many researchers want to study whether specific effects observed in human reasoning arise

Figure 1: **ReCogLab** is a customizable framework for generating relational reasoning evaluation examples. We show example problems and graph structure for Comparison, Social Networks, Syllogisms and JSON Families reasoning tasks. These tasks require a variety of important relational reasoning capabilities such as transitive inference, working memory capacity, and set reasoning. Our framework's customization allows us to probe language models on increasingly difficult classes of relational reasoning problems generated by our test configurations. See Appendix D for generated examples on each of these tasks.

in LLMs, such as content sensitivity (Lampinen et al., 2024). Often researchers will have to create specific datasets by hand to probe these effects because fixed benchmarks do not have the flexibility to measure the parameters of interest.

We aim to solve these problems with a generative approach by introducing a framework which creates datasets that can be used for different purposes. We introduce ReCogLab (see Figure 1) for generating relational reasoning problems over long contexts that can be customized for many purposes. The key aspect of this dataset is virtually every aspect of the generating process can be controlled via a configuration file specifying parameters. For instance, in all tasks **we can change the complexity of the reasoning problem**, allowing for the difficulty of the problem to scale up as models become more powerful or scale down for diagnosing poor performing models. The fine control of parameters such as the ordering of statements, the complexity of the language, or the amount of filler context **allow us to probe models** behaviors along many different dimensions. Finally, because of its configurability, ReCogLab can be adapted for use by both ML and Cognitive Science community to easily run a wide variety of exploratory experiments.

With ReCogLab, we perform a number of experiments and probes to demonstrate how our framework can be used to scale to model performance, probe and analyze model performance, and recreate Cognitive Science experiments on humans and language models using our framework like ordering and congruency. Using our framework, we can identify several language model performance characteristics on different relational reasoning tasks. Our framework allows us to isolate failure breakpoints and provide insight into understanding general relational reasoning capabilities in language models.

In this paper we: (1) introduce ReCogLab, a new flexible dataset framework for scalable evaluation and cognitive science-inspired probes of LLMs on relational memory (2) perform experiments showing how our dataset can scale and adapt over time to remain a useful evaluation framework (3) show how our configability can be used to probe and analyze different problems in relational memory (4) study several observed and hypothesized relational reasoning phenomena reported from the cognitive science literature enabled by the procedural generation of structured, text-based prompts. We release the ReCogLab framework and datasets used in this manuscript as a useful tool for evaluating examining LLMs for relational reasoning.

## 2 BACKGROUND

**Relational Memory and Reasoning in Cognitive Science.** Relational memory and reasoning have been studied by psychologists across a variety of domains (Behrens et al., 2018; Eichenbaum, 2004; Nelli et al., 2023; Kumaran & Maguire, 2005; Tolman, 1948; Stupple & Ball, 2008; Evans et al., 1983). A common experimental probe is "transitive inference" (Burt et al., 1911; Piaget, 1957): subjects are given data with the form "A>B, B>C, C>D, D>E" and then queried about relationships that they did not observe directly but can be inferred (e.g. B>D). Different experiments vary parameters of transitive inference to construct meaningful insights into cognitive processes. We draw inspiration from this literature, measuring effects of presentation order (Hotta et al., 2015) and symbolic distance (Moyer & Bayer, 1976) with 8 different language models. We also explore how these models' relational reasoning abilities differ in sequential processing versus feature learning settings (Koster et al., 2018) through our congruency experiments.

Other experiments have studied humans ability to reason about and sometimes navigate more complex relational structures, such as trees, grids, or community graph structures (Mark et al., 2020; Schapiro et al., 2013; Garvert et al., 2017; Lynn & Bassett, 2020). These experiments inspired our more richly structured, configurable options in our framework. Relevant work in psychology has also studied syllogistic reasoning (Chater & Oaksford, 1999), and connections have been hypothesized that link syllogistic reasoning with reasoning over transitive chains (Guyote & Sternberg, 1981). Indeed, a notable failure mode is that humans appear to simplify syllogisms to create easier-to-reason-about transitive chains (Ceraso & Provitera, 1971), which informs our experiments on symbolic distance.

**Evaluation of LLMs** Early work on reasoning benchmarks for language models[1] includes bAbI (Weston et al., 2016), a QA-style synthetic benchmark requiring very simple language understanding and reasoning questions, and Knowledge Base Question Answer (KBQA) Berant et al. (2013); Lopez et al. (2013), a system for evaluating multi-hop reasoning on knowledge graphs. Like our work, KBQA is heavily inspired by using graphs to assess relational reasoning capabilities.

More recently, datasets such as BigTOM (Gandhi et al., 2023), ToMi (Le et al., 2019) and Rule-Takers (Clark et al., 2020a) have been developed requiring much more complex reasoning. Other relevant work has investigated logic programming for logical deduction (Li et al., 2024c), multihop QA (e.g. (Li et al., 2024b)), and knowledge graph reasoning (e.g. (Luo et al., 2023)).

Other related work includes long-context LLM evaluations (Levy et al., 2024; Bai et al., 2024; Vodrahalli et al., 2024). Li et al. (2024a) in particular includes some of the aspects of relational reasoning but without our dataset's customizability. All of these works focus specifically on this question of thoroughly examining the context length of language models. Our dataset is unique in affording a joint inspection of context length in the context of relational reasoning, and affords scaling complexity along other directions with customizabilty along multiple dimensions.

**Comparison of LLMs to humans** Substantial work has compared reasoning in LLMs to humans (Hagendorff et al., 2024; Binz & Schulz, 2023), employing cognitive science and psychological methodologies to explore the biases in LLMs Jones & Steinhardt (2022); Seals & Shalin (2023); Webb et al. (2023); Berglund et al. (2023). Some of the cognitive effects that we study here have been studied individually in some of these works such as: consistency of logical reasoning with prior knowledge Lampinen et al. (2024), syllogistic reasoning Eisape et al. (2024), premise order in syllogisms Chen et al. (2024), and susceptibility to irrelevant information (an aspect of capacity experiments in. We similarly adopt experiments from the cognitive science literature as a reference point for understanding LLM behaviors, but in the context of relational memory and reasoning. We show many of these experiments in Section 4.3.

**Sandbox Evaluation Toolkits** We have proposed a framework for automatically generating new templated datasets for relational reasoning problems. Other works have looked to increase the speed of creating evaluations, for instance Dynabench Kiela et al. (2021) and later Dynatask Thrush et al. (2022) which allow for rapid integration of models into dataset creation. However, this still requires human annotation and thus would not be able to automatically change parameters of the dataset instantaneously as our framework does.

Cognitive scientists have long been interested in evaluating LLMs as we do with humans Hernández-Orallo (2017) and environments such as animal-ai environment Voudouris et al. (2022) have been created. However, this environment is a 3D embodied environment, and while it also touches on memory, it does not touch on language or on relational reasoning specifically. Another close comparison is CLEVR Johnson et al. (2017), a dataset of synthetically rendered 3D scenes that can be used to query for simple relational reasoning capabilities from images. Our work extends upon synthetically templates scenes by using a generative framework and language to precisely construct more complex relational reasoning examples.

## 3 RECOGLAB

Unlike a dataset which consists of static evaluation examples, the ReCogLab framework is a generative system that deterministically creates a relational reasoning dataset based on a particular input

---

[1] As well as other AI systems, see e.g. Cropper & Dumančić (2022)

configuration and seed. Our highly customizable system configuration has many tunable knobs required to support the breadth of experiments needed to explore relational reasoning.

This key difference gives us several advantages when running experiments.

- **Effect Isolation:** We can design experiments to isolate specific reasoning biases in language models that are well supported in human cognition studies. Not only can we show the existence of these biases, we can also measure their effect on problem-solving.

- **Safeguards against leaks:** With the advent of internet-scale training data, there are high risks of accidentally leaking dataset examples into training data. Not only is this hard to detect, if it does happen it becomes impossible to compare numbers before and after the leak. A generative framework avoids this by providing instructions for re-generating the same configuration of the dataset but with completely different test seed. We show that despite re-seeding a configuration, model performance across different seeds are statistically consistent under different experiment configurations.

- **Scalable Difficulties:** A framework enables the creation of datasets with arbitrary difficulty, allowing for the measurement of language model capabilities across various models, from compact device models to large, state-of-the-art models.

## 3.1 GENERATIVE FRAMEWORK

Examples generated by ReCogLab come from one of several relational reasoning tasks: **Comparison**, **Social Networks**, **Syllogisms** and **JSON Families**. See Appendix B.2 for more details and results. We chose these tasks because they can be generated automatically and include a spectrum of problems in relational reasoning. Importantly, each of these tasks can be characterized as reasoning about graph problems using language. See Figure 1 for examples from each dataset and the corresponding logical graphs associated with each.

Each example consists of a context $C$ - multiple factual statements that the model assumes to be true (e.g. "Miranda is related to Steve"). Each of these facts denotes a relationship between pairs of entities $(E_i, E_j)$ (a person or object or other noun in the relationship) which compose edges within a graph. Next the question $Q$ which relies on a subset of the context $C$ to answer, requires integrating information across distinct facts to unambiguously produce an answer or answers $A$. We emphasize that ReCogLab is not just a dataset, but a framework for automatically generating infinite datasets and dataset examples. Our framework can control the core difficulty of the graph problem by increasing the number of nodes, the number of relationships, and even the structure of the graph. In addition, we can introduce auxiliary difficulties to the graph problem such as using "flavor text" or scrambling the order of statements in the context. Because we control the data generating process by specifying parameters, we can produce datasets that isolate specific auxiliary and core difficulties to understand how they affect the capabilities of different language models. Please see Appendix B.1 for more details about how each task procedurally generates test examples.

## 3.2 TASK DOMAINS

**Comparison.** A comparison is a directed edge in an acyclic graph where vertices are *entities* and edges describe a directional comparison between them. Inspired by transitive inference, we construct comparison problems using three attributes that exhibit transitivity: comparison of age, size, or weight of objects.

For comparison reasoning problems, we target specific parameters to understand how language models behave when presented with similar problems in different forms such as: the comparison type, the number of objects, the ordering of comparisons, and the attribute types of directional reasoning.

**Social Network.** Inspired by Kumaran & Maguire (2005), the next set of tasks we propose requires reasoning about Social Networks. Like comparisons, a social network can be algorithmically represented as a graph problem where nodes represent people and edges represent a particular relationship like friendship. Unlike with Comparisons where relationships are directional, Social Network assumes relationships are symmetric and therefore graphs are undirected.

A social network is good for probing long-context relational reasoning for several reasons. First, social networks can be arbitrarily structured. This makes it possible to generate very large graphs

with complex structure. Secondly we can introduce "flavor text" (see Appendix B for more details) generation in every example's *context* to provide a more story-like problem. Finally one could incorporate other kinds of social network edges like teacher-student (directed edge) or even enemies (undirected edge with a different semantic interpretation) to build even more complex reasoning probes.

**Syllogisms.** Previous work on syllogistic reasoning has shown interesting parallels between humans and language models (Eisape et al., 2024). We extend prior work that has studied ordering and congruence effects on humans and language models to include a capacity dimension. We construct chains of propositions that transitively resolve to a single final conclusion and prompt the language model to choose between all possible conclusions with the given subject and predicate.

**JSON Families.** Inspired by knowledge based question answering (Lan et al., 2021) on structured problems, we also consider how relational reasoning is impacted by the use of programming language input like JSON. We created a generating process for document graphs where elements in the JSON are nodes in the graph that connect to the parent object. We use this to model knowledge and generate testable questions like whether two individuals have the same hobbies.

### 3.3 DATASET CREATION

For each task, we create different configurations that generate datasets used in this paper with fixed seeds to guarantee consistency and reproducibility. We generate sweeps of different configurations targeting evaluations of a specific cognitive probe. We generate 50 validation examples and report on 250-1000 test examples. Here the important thing is not the specific number of examples, but the ability to tailor skill and difficulty levels to measure cognitive abilities in different LLMs. We provide technical details in Appendix B and open-source code[2] for generating examples from ReCogLab.

## 4 EXPERIMENTS

Here we demonstrate the flexibility in benchmarking, probing and experimentation of the ReCogLab framework. We find that our generative approach is not hindered by sampling issues by showing that different seeds generate comparable evaluations in expectation.

Next, we look at how our framework can be used to isolate critical problem parameters on reasoning performance and the relative effect of increasing problem complexity in Section 4.1. We also investigate whether the exact way in which we ask the task or the reasoning complexity of the language affects performance in Section 4.2.

Finally, we show the flexibility of our framework to reproduce Cognitive Science inspired experiments such as congruency, ordering effects, premise consistency, and indeterminacy in Section 4.3.

**Language Models** We evaluate several open-source and closed-sourced models on our suite of generative reasoning benchmarks. In our results, we show Google's open-source Gemma-2B, 9B, 27B family in green. The open-source Mixtral models are shown in orange. Gemini Flash and Gemini Pro (closed-source) are shown in blue. OpenAI's closed source GPT-4o is shown in red. Each specific probe involves validating on 50 examples before selecting the best prompt and parserfor test-time evaluation on. Please see Appendix A for the library of prompts and parsing strategies.

**Seeded Sample Variance Analysis** When using a framework to evaluate capabilities, the test configuration defines a shareable benchmark to evaluate multiple models against. However, because a benchmark is constructed by randomly generating examples from a framework given a configuration, there exists sample variance in benchmark scores. Our key insight is that ReCogLab generates examples based on a seed and configuration file in such a way that different seeds produce the same dataset "in expectation."

To understand the behavior of a generative framework's sampling variability, we run several "re-seeding" experiments. For 8 different configurations on Social Network, we generated 30 different seeded datasets of 200 examples each. This gives us 30 re-seeded accuracy scores of a single configuration for every model we benchmark on. Because measuring sample variance is computationally expensive ($30 * 8 * 200$ test examples), we restrict the analysis to Gemma-2B, Gemini Flash and

---

[2]https://github.com/google-deepmind/recoglab

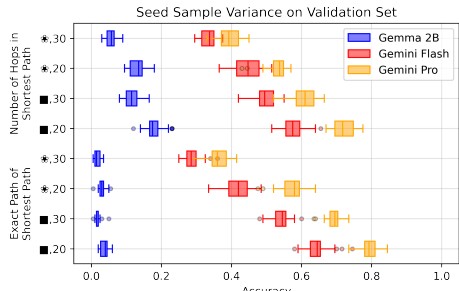

Figure 2: **Re-seeded Variance**. We show the distribution of 8 re-seeded config experiments in a boxplot and find that regenerated datasets from the same config score similarly. ❊ and ■ are tasks with and without *flavor text* enabled and the number of entities used during generation. This shows that ReCogLab results are reproducible under a new seed.

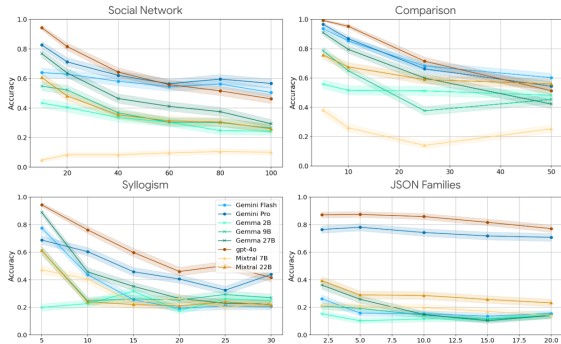

Figure 3: **Capacity vs. Accuracy**. We evaluate each language model's long context capacity by increasing the number of entities for each of our tasks. We find in all cases that performance drops, although this drop is less severe in more capable models such as GPT-4o and Gemini Pro

Gemini Pro. We measure performance on a single chain-of-thought prompt which has good performance across all model families we validated on. Our results in Figure 2 indicate that performance between different seeds of the same configuration are comparable and form valid hypothesis testing distributions. While it would be rigorous to demonstrate these properties on all evaluations presented in this paper (and future benchmarks configurations too), we posit that these guarantees hold true based on Figure 2. Therefore future evaluations using the same configuration are comparable even with different seeds and we can always reproduce results from a single-seeded experiment.

## 4.1 SCALING EXPERIMENTS

**Capacity.** One application of ReCogLab is generating long-context reasoning problems automatically for evaluation or training. Context can be added in several different ways such as the total length, number of relations, or the size of irrelevant context. In Figure 3, we evaluate how reasoning ability depends on the number of entities for each task domain by problems iwth different graph sizes. We find, perhaps unsurprisingly, that performance generally degrades as the complexity of the problem increases across all models. The more capable closed-source models (Gemini Flash, Gemini Pro, GPT-4o) perform the best.

We also have other studies of reasoning performance and capacity including slicing by symbolic distance and slicing by the type of irrelevant content being added in subsequent sections.

**Symbolic Distance.** Symbolic Distance, or the number of relations that separate two entities in a graph, is understood to be a key parameter in relational reasoning performance(Moyer & Bayer, 1976). A symbolic distance of 1 implies there is exactly a one statement in the input that gives sufficient information to answer the question, in which case the query reduces to a Needle-in-Haystack (NIH) probe. In Figure 4, the first row of each heatmap corresponds to a NIH evaluation on each model which shows minimal degradation across all long context sizes. ReCogLab enables a more robust analysis of symbolic distance.

We find that most models have strong performance when only 10 entities are involved in the graph, regardless of the symbolic distance. However as the graph size grows, models ability to reason about the relationship shrinks to only reliably answering correctly at a symbolic distance of 6. With Gemini Pro and GPT-4o, we also evaluate up to 70 entities and find that the quality of the prediction stabilizes, in other words adding more entities doesn't degrade relational reasoning capabilities as long as the number of traversals required is less than 6. For more detailed commentary and analysis on other tasks see Appendix C.1.

## 4.2 QUESTION TYPE AND LANGUAGE EFFECTS

Next, we examine the capability for ReCogLab to further tease apart how a model performs on these problems. We want not only to show the relative performance of different models on a task; we want to understand the effect of question and language degrade different model's capabilities. A key aspect

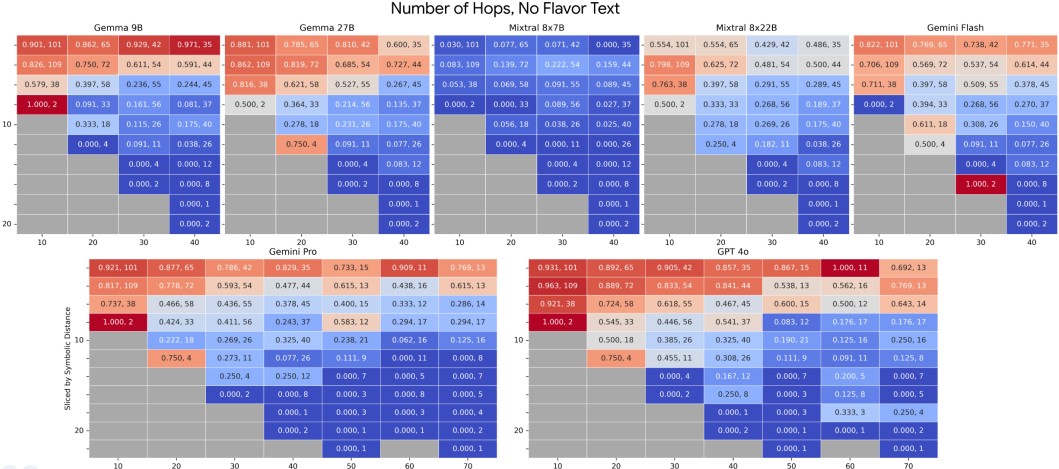

Figure 4: **Symbolic Distance and Graph Size**. Here we show the each model's performance on a Social Network task, sliced by the Symbolic Distance and Graph Size. The x-axis denotes the graph size in terms of number of entities and the y-axis denotes the symbolic distance required to answer. Each box reports the accuracy and number of test examples in the test set in the slice. Red indicate higher accuracy on a slice. See Appendix for this analysis presented on other tasks.

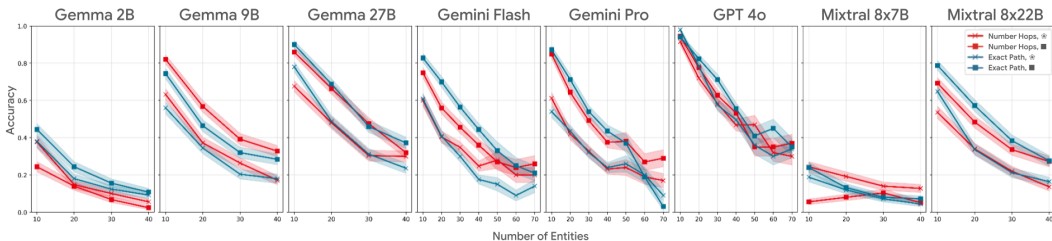

Figure 5: **Social Network Reasoning**. Here we show two tasks and two ablations on all models. The two tasks consists of identifying the **number of hops** and the **exact path** of a shortest path in a Social Network example. See Section 4.2 for deeper discussion of the results. The x and square ticks are benchmarks with and without flavor text respectively.

of ReCogLab is the ability to change different aspects of the dataset easily. Figure 5 shows the Social Network problem with various different perturbations.

For each graph problem generated by Social Network, we ask two questions. The first is, given two nodes of the graph, what is the **Number Hops** of the shortest path from one to the other. The second is a harder version of the problem, what is the **Exact Path** of the shortest path. For **Exact Path**, we also include instructions to format list answers as a Python code block since all models demonstrate strong proficiency in generating Python code. We ablate several factors among graph problem such as the use of flavor text, and the question type in Figure 5.

Flavor text is a problem augmentation where the presentation of the graph information becomes more complex. This has two effects, increasing the number of input tokens while decreasing the information density per token. We observe in Figure 5 that in the majority of models, flavor text obstructs reasoning performance. Barring GPT-4o, this difference suggests that language of information has a critical role in solving the problem.

We also explored difference between the same context asked with a different question per example. Exact Path is a strictly harder problem than Number Hops, the solution to Exact Path also solves Number Hops. However many language models, Gemini Pro, Gemini Flash, and Mixtral-7x22B show better performance on the harder problem Exact Path. An initial explanation for this behavior lie in the number of decoded tokens which is higher for Exact Path.

**Complexity and Filler Text.** Another question that comes up when we examine the scaling results of Figure 3 is what is actually making the problem difficulty scale. There are two separate things happening. In Social Network, as we add entities/nodes to the graph, we not only make the reasoning problem more difficult because the model has to make more logical deductions, we also increase the

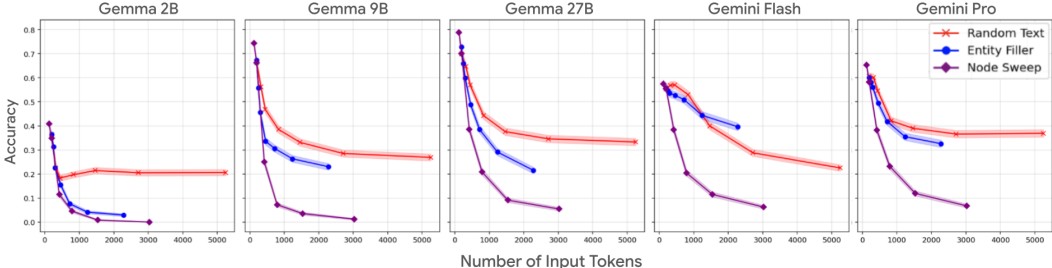

Figure 6: **Filler text.** Performance of different models using different kinds of filler text to increase context length. *Random text* fills context with public domain books, *entity filler* adds distractor sentences about entities but do not effect the answer, and *node sweep* is our base experiment with an increasing number of entities.

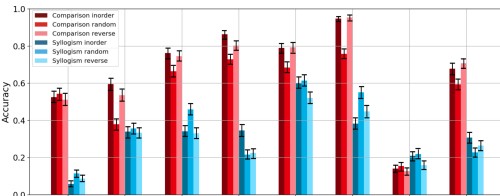

Figure 7: **Transitive Inference and Context Ordering**. We evaluate the effect of context ordering on transitive inference performance. Similar to patterns observed in human cognition, when the context is presented in topological order, all language models improve in relational reasoning capabilities over random. Even reverse topological order improves results as the statements are still in some sorted order.

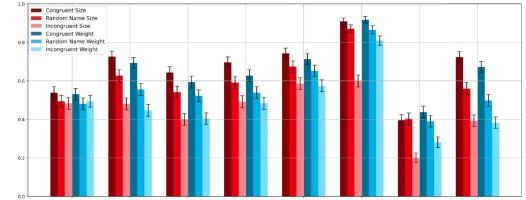

Figure 8: **Congruent and Incongruent reasoning.** We find that transitive inference is strongly impacted by real-world priors. Across different language models, we observe degradation in reasoning performance when presented with logically consistent, but incongruent statements. The *Random Name* ablation removes any connection between the graph problem and real world priors as a control.

context length of the problem which makes it more difficult for the model to identify the important context.

To further analyze this, we do an experiment where we separate out these two effects. We take the **Number Hops** task from Social Network and show the effect of adding additional context to several of the models. In the first version, we simply increase the number of entities, increasing both the complexity of the problem and increasing the context length. In the second version, we add more distractor "flavor filler text" about the entities that are irrelevant to the question. In the final version we add completely irrelevant text like lines from public domain novels to create more input context.

In Figure 6 we plot accuracy as a function of the number of input tokens for an even comparison across these different ways of increasing context length. We can see pretty clearly that adding graph complexity is by far the hardest for models than either kind of filler. Interestingly, we do see that there is a real drop in performance when we add *any* extra tokens, including filler, but the drop off is much less steep. We also note that in several of the models such as Gemini Pro, the random filler context degrades performance up to a certain point before plateauing, suggesting that needle-in-the-haystack type filler problems have a limit to how much they can make problems harder by simply adding irrelevant context.

## 4.3 COGSCI INSPIRED EXPERIMENTS

**Ordering Bias.** Transitive inference is possible for humans even when the temporal order differs from the symbolic ordering (e.g. "$B > C$ and $A > B$"). However, order affects sequence memory in humans (Domjan, 2010), and when the temporal order and symbolic order match, performance should in principle improve. Thus, we wanted to evaluate our models' ability to reason accurately about associations presented in context, and measure whether there was a dependence on presentation order in their context across different domains.

In Figure 7, we consider several ordering experiments on Comparison and Syllogisms. In particular, we construct linear chain of directed entities and order the context according to the linear chains' topological order. This means that adjacent sentences always include at least one entity in common.

For Comparison, we can clearly see across nearly all models that randomizing order causes a noticeable degradation. This implies that topologically sorting statements can improve relational reasoning performance on the same underlying problem.

**Congruent and incongruent.**   Despite only being trained on textual data, language models exhibit remarkable knowledge about real world physical attributes of objects. This can be easily demonstrated when asking questions like: *Are fire trucks larger than teddy bears?*. This prior knowledge is aligned with facts found in the training data and is essential to many applications of language models.

Motivated by the findings of Lampinen et al. (2024), who report that logical reasoning is more accurate when logical premises are congruent with real-world facts for LLMs (as it is for humans), we explore this in the relational reasoning regime. Expanding on their paradigm, we categorize premises based on their grounding in real-world knowledge. *Congruent* relationships are those that are consistent with the real world (e.g "whales are larger than minnows"). Conversely, *incongruent* relationships contradict our prior knowledge about the world ("minnows are larger than whales"). Finally, we consider a third category of relationships involving contrived, nonsensical objects ("3x0Bn are larger than hJp65") that have no meaningful semantics behind the name. Intuitively, congruent relationships are more likely to be consistent with the prior training of the language model and may bias the answering of these questions.

First, we investigate the impact of congruence on language model performance, replicating the observation that congruence contributes positively to performance.

For the Comparison task, to extract real-world knowledge about objects, we curated a list of 540 objects and asked Gemini Pro to estimate their size in kilograms and meters. From this, we construct comparison problems that are **Congruent** and **Incongruent**. We also generated a **Random-String** version of the comparison problem which replaces all object names with a random string of alphanumeric characters of length 5 for the contrived, nonsensical baseline.

We find that using **Congruent** statements outperforms similar comparison problems constructed with **Incongruent** statements. This behavior is consistent across language models and relational reasoning tasks. Some models even exhibit worse-than-chance (0.5) correctness on **Incongruent** comparison problems. **Random-String** performance lands between the two extremes, suggesting a positive relationship between reasoning competency and factual coherence.

**Identifying Inconsistent Premises.**   One important capability of a reasoning system is identifying when a premise is logically inconsistent (Black et al., 1986; Johnson-Laird et al., 2004). This reasoning task is different from the previous structured reasoning tasks as identifying the set of inconsistent facts requires global reasoning capabilities. Furthermore this question does not incorporate any specific entity name so there's no prior on which statements from the context are relevant; and there are no rules on organizing or processing statements that trivialize this task. Therefore all statements must be equally considered before a conclusion of logically consistent can be drawn.

Here, a logically consistent directed graph assumes no cycles exist. If A > B, B > C, then you cannot have C > A. This implies that all inconsistent graphs are comparison graphs that contain a cycle. To create inconsistent examples, we start with a valid comparison directed acyclic graph (DAG) and randomly sample nodes to add an edge that would induce a cycle and turn the problem inconsistent.

We show the confusion matrix on predictions extracted from each LLM on the first row of Figure 9. Comparison across diagonal elements indicate that larger models perform better on detecting inconsistent statements. We provide deeper analysis on the prediction calibration issues in Appendix C.2.

**Indeterminate reasoning.**   Another important aspect of a reasoning is detecting when there's insufficient information to draw a conclusion. This relates to an ability often ascribed somewhat uniquely to humans: our ability to characterize uncertainty (Courville et al., 2006). Being able to detect *indeterminacy* is important for several reasons. First, a reasoning system that fails to identify an indeterminate premise may draw erroneous conclusions that are not supported by the evidence.

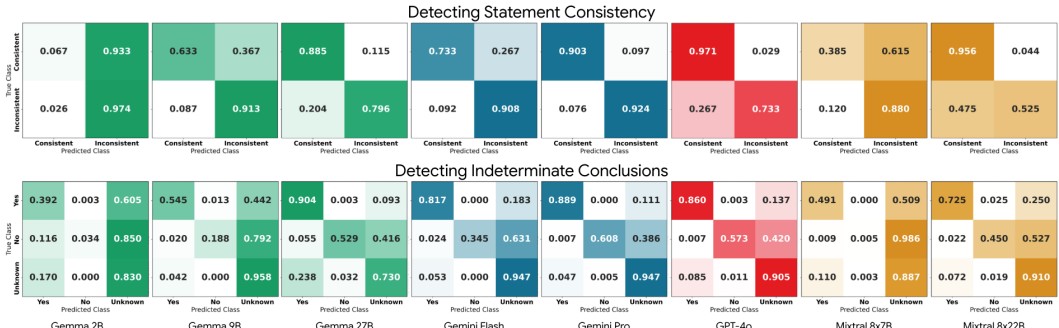

Figure 9: **Confusion matrix on meta-reasoning**. ReCogLab allows us to probe language models for important meta-reasoning capabilities. The first row shows confusion matrices on classifying *consistent* and *inconsistent* premises. We find later generations of models iterations can better identify statements that are inconsistent. We also find a similar trend when inferring whether a conclusion is *indeterminate*. These are important meta-reasoning capabilities with critical implications for designing human interaction and autonomous decision-making systems.

Second, recognizing when a conclusion cannot be drawn is important for improving decision making and handling ambiguity.

We start by generating a random DAG tree of comparison problems. Under a DAG tree, querying a comparison between two nodes may result in an indeterminate conclusion if the two nodes no longer have a path to each other. In other words, insufficient *context* was provided to reason about the relationship between the two. This results in the modified answer set consisting of three potential answers (Yes, No, Unknown). For this particular evaluation, we provide instructions to answer whether the context and question are indeterminate before asking a question. While this experiment follows by combining cognitive science experiments about uncertainty estimation and transitive inference, it has not to our knowledge been performed in the cognitive science literature.

We show results of this probe in the second row of confusion matrices in Fig. 9. Similar to the previous probe, the diagonals are correctly classified examples. We also analyze calibration biases in Appendix C.2 and find that many language models disproportionately prefer to answer *Yes*. This has strong parallels to "acquiescence bias" towards answering affirmative that has been documented in humans in a variety of settings (Krosnick, 1999).

## 5 DISCUSSION

We introduce ReCogLab, a flexible, generative framework for quantitatively measuring relational reasoning abilities in LLMs. We demonstrate the utility of this approach, conducting a number of experiments to benchmark relational reasoning performance across different models and problem complexities. Moreover, we recreate cognitive science experiments that characterize how reasoning performance depends on features such as presentation order (Hotta et al., 2015), symbolic distance (Moyer & Bayer, 1976), and congruency with prior experience (Lampinen et al., 2024). We also create a number of novel experiments that measure how models recognize inconsistencies and indeterminacies, an important contribution for model safety. Ultimately, we hope that by accurately measuring the cognitive capabilities of our language models, we can contribute a metric for hill-climbing improvements in the field that are inspired by human cognition. Furthermore, we hope that this work will inspire deeper, rigorous probes into reasoning capabilities of both humans and artificial agents, and help provide insight into the success and diagnosing failure modes of reasoning.

Many of the effects we observe in LLMs mirror effects observed in humans, suggesting commonalities in the factors contributing to relational reasoning in LLMs and humans. While it is instructive to compare and contrast LLM reasoning to that of humans, we should as always be careful about drawing too many conclusions about the similarities between LLMs and human psychology (Shevlin & Halina, 2019; Shanahan, 2022). Even when there are similarities, there is ambiguity about the explanation: it may derive from similarities in the statistics of experienced data, task objectives, architectures, or learning rules. Alternatively, it could be that two entirely different mechanisms are responsible for the same effect.

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

## A PROMPT TEMPLATE VALIDATION AND MODEL CAPABILITIES

While prompt strategies are an important line of research for exploring language model capabilities, they introduce additional variance to our observations of a language model's true problem-solving capabilities. From a scientific point-of-view, this is problematic; optimizing prompt templates (even if they are 0-shot) for specific tasks may yield better hill-climbing results, but they make it harder to transfer our understanding of a language model's general capabilities to a different task.

To mitigate this issue, we treat prompts template and answer parsing as hyper-parameters to fit on a validation set first. Because ReCogLabis a generative framework, we can generate always generate validation sets to find the best prompt template strategy for each and every language model. This means that every language model is given the opportunity to figuratively "play to their strength."

In our experiments we only consider "0-turn" and "0-trained" prompt strategies to preserve the generalization of our results. We also use several prompts that targeted specific sub-datasets or tasks like Comparisons, Syllogism, and Feasibility because their setup requires more explanations. We show the prompts we tested below. [question] refers to both the contextual information and task-specific question combined.

Common Prompts Templates used in all tasks.

- [question]
- [question]
  Answer in only one word.
- [question]
  Think through your answer then respond at the end with a newline and 'Answer:' with your answer.
- [question]
  Think through your answer then respond at the end with a newline and 'Answer:' with your answer. Use only one word for the answer.
- [question]
  Let's think step-by-step

Social Network Prompts Templates

- You are a language model with advanced cognitive abilities. Your task is to understand and reason about the following social scenario, much like a human would. Read the story carefully and answer the questions that follow.
  [question]

Comparison Prompts Templates

- [question]
  Answer the above relational reasoning question with Yes or No. Use only one word for the answer.
- You are a language model being probed for your reasoning abilities. Your task is to carefully think about the following information and answer the question.
  [question]
  Make sure to respond at the end with 'Answer:'
- [question]
  Answer the above relational reasoning question with Yes or No with a newline and 'Answer:' with your answer. Give your best guess if uncertain. Use only one word for the answer.
- [question]
  Answer the above relational reasoning question with only Yes or No with a newline and 'Answer:'. Give your best guess if uncertain. Use only one word for the answer.
- [question]
  Answer the above relational reasoning question with Yes, No, or Unknown. Use Unknown if the question cannot be answer with the information given. Use only one word for the answer.

We find that after validation, different models from the same family of model classes perform better with different prompts.

This framework of splitting prompts and model capabilities works for more complex patterns of cognition like Chain-of-Thought and In-context learning. We also consider sentence parsing but that has a much weaker impact on performance and is also significantly cheaper to test since they re-use the same language model base prediction. We will release the prompt and sentence parsing functions when releasing code.

## B  FRAMEWORK DETAILS

We discuss further details about ReCogLab.

### B.1  GENERAL OVERVIEW OF FRAMEWORK

Here we explicitly describe how the framework constructs training examples to support different experimental hypotheses. We will make the entire framework codebase and dataset available to the public.

Our framework works by:

1. Creating a config file for the dataset you want to create. This includes the task and sub-task (see the longer task descriptions in Section B.2) and the task parameters (see Section B.3).

2. The user passes in this config as well as the dataset split and the random seed to the dataset generator class.

3. Based on which task is chosen, the dataset generator for the corresponding task generates the underlying logic of the graph problem (e.g. for Social Networks it generates an undirected graph) and generates the context $C$ text, question $Q$ and answer(s) $A$ from templates. The context is made up of a list of relationship $R_{ij}$ between pairs of entities $(E_i, E_j)$ (each of which is a person or object or other noun in the relationship or can be an attribute). The templates then fill in the text to describe the relationship $R_{ij}$ (often sampling from a number of possible templates) and sampling entities $(E_i, E_j)$ to fill in. The generator also enforces and uses all of the task parameters to do this generation. For details on each of these constructions see the corresponding paragraph of Section B.5.

4. The final example is then saved and the generator continues to generate up until the desired dataset size $N$.

5. For experiments which "sweep" parameters, the dataset generation of $N$ examples is repeated, changing the value of the sweep parameter on each step to generate all $C_P$ parameter configurations.

### B.2  TASK DESCRIPTION

- Comparison
    - Older-Younger: Constructs comparisons on age between people.
    - Larger-Smaller: Constructs comparisons on size. These can incorporate congruent/incongruent knowledge priors.
    - Heavier-Lighter: Constructs comparisons on weight. These can incorporate congruent/incongruent knowledge priors
    - Consistency Detection: Instead of asking a comparison question, asks whether the statements are logically consistent.
    - Indeterminate Conclusion: Generate a comparison tree and ask whether the statements support drawing a conclusion.
- Social Network
    - Fastest Message: Given the goal of passing a message from person A to person B with the fewest hops, who should person B give a message to. Reduces to a logical breadth first search.

- Oldest/Youngest Generation: Given a statement about a family, who is the oldest and youngest generation? Similar to Older-Younger.

- Syllogism

  - Set Membership: Given statements describing group membership between different labels, determine how two labels' members intersect.

- JSON Families

  - Family Size: Calculate how many members are in a specific family.
  - Family Member Hobby: Given a hobby, check whether it is a hobby of a specific family member.
  - Family Size Comparison: Given two families, compare their size.
  - Hobby Comparison: Given two specific individuals in different family, describe overlapping hobbies.
  - Age Comparison: Given two specific individuals in different family, compare age.

## B.3 TASK PARAMETERS

- network_type: Whether to construct the task using a linear chain or a randomly generated tree.

- num_entities: The number of entities used when generating an experiment. We use this for evaluating capacity performance.

- entity_type: The type of entities we construct word problems out of. These consist of names, objects, labels depending on the need of the task.

- congruence: If the task supports integrating real world priors to generate congruent or incongruent statements.

- randomize_relations: For a directional statement, whether to reverse the statement and relation. $A > B$ becomes $B < A$ in the word problem.

- do_reverse_comps: To evaluate whether the order of the statement in the question matters.

- relation_type: For social network, applies labels to the edge which include other kinds of relationships.

- num_families: Equivalent to num_entities, measure of capacity for length (total number of families).

- num_family_members: Equivalent to num_entities, measure of capacity for width (max members within a family).

- hop_length: Distance between two families chosen. If not set, default is sampled randomly between half the number of families to overall number of families.

## B.4 COMMON TECHNICAL DETAILS

We use NetworkX Python library to construct randomly generated graphs. To seed deterministic randomness, we use Jax's PRNG Keys to ensure that each example is isolated from each other while still being fully reproducible given the correct key. We used Jax v0.4.33 randomization implementation since the PRNG key behavior is consistent within a version.

Our experiments generate linear chain graphs or random tree graphs. A chain graph is a sequence of nodes where each node is connected to the next one in a linear sequence. This allows us to test for ordering effects in the context because answering the question of a linear chain graph is equivalent to traversing the statements sequentially. Other experiments generate more complex graphs like random trees.

We incorporate multiple stages of deterministic randomness, first at initialization to help randomly configure parameters shared across all test examples, and next at a per-test example generation.

## B.5 DOMAIN-SPECIFIC DETAILS

**Comparison**  We use NetworkX to generate linear chain and random trees depending on the sub-task.

Comparison sub-tasks consist of asking questions about a bunch of entities in three comparative settings, *Size*, *Age*, and *Weight*. *Consistency Detection* and *Indeterminate Conclusion* use the *Age* comparison but ask a modified question.

For evaluating the ordering effects, we use a linear chain graph shown in the contextual premise ordering experiments in 7. Here inorder and reverse corresponds to matching the toplogical sequential order of the linear chain. Random simply randomizes the ordering. We also use linear chain for Congruent and Incongruent reasoning as well as the Symbolic Distance probe as they simplify the problem construction.

For indeterminate reasoning and inconsistent detection, we use random trees. This is because creating an inconsistent premise requires making a self-loop which is trivial to identify in a linear chain. It's also impossible to construct unknown or infeasible questions with a linear chain because every pair of object has a relationship.

**Social Network**  We use NetworkX to generate linear chain graphs for evaluating capacity.

We evaluate on the sub-task of **Fastest Message** which is a specific question that asks who should [Entity 1]pass a message to have it arrive to [Entity2]in the fewest hops.

To construct a **Fastest Message** problem, we generate premises about the nodes in the graph where edges denote a friendship and a way for a message to be passed. We randomly pick two entities and ask how to pass a message. Here the answer can be determined through breadth first search.

**Oldest/Youngest** sub-task reformulates the social network graph to a directed problem where nodes indicate relationships between parent and child. The **oldest/youngest** generation asks which family member is part of the oldest or youngest generation. We can solve and verify the answer using a combination of common-ancestor graph algorithms. While we didn't report benchmark performance on this task, we note that **oldest/youngest** has several interesting, distinguishing properties from other tasks presented in this paper. First it does not incorporate any entity names in the question which might inform the language model how to search the context. It also requires processing every bit of information before being able to verifiably prove the answer.

**Syllogism**  We use custom-written logic to generate syllogisms with an arbitrary number of propositions. For a given number of propositions, we keep track of the current valid conclusion, and enumerate all new propositions which, along with the current conclusion, generate a new valid conclusion. This is done in a depth-first way, and when we hit the desired number of propositions, we yield the current chain.

We evaluate the model by showing it all of the propositions in the chain, as well as all syllogism types with the valid conclusion's subject and predicate, i.e. "All A are Z," "No A are Z," "Some A are Z," and "Some A are not Z."

To study ordering effects, we sort the propositions by constructing a Hamiltonian path between the proposition which contains the subject of the conclusion, and the proposition which contains the predicate of the conclusion.

**JSON Families**  Inspired by knowledge based question answering (Lan et al., 2021) on structured problems, we also consider how relational reasoning is impacted by the use of programming language input like JSON. We created a generating process for document graphs where elements in the JSON are nodes in the graph that connect to the parent object. We use this to model knowledge and generate testable questions like whether two individuals have the same hobbies.

For the JSON Families task, our framework produces structured, nested JSON representations of multiple families. Each family is identified by a last name, address, and a set of members. Each member is then identified by their name, age, and hobbies. Table 1 outlines the kind of questions i.e. probes that can be be conducted on this dataset. The question types range from simple fact retrieval (e.g., family size) to complex comparisons (e.g., relative age of members across different families or

shared hobbies). This structured data allows us to easily scale the number of families as well as the members per family.

During the dataset generation process, we ensure that the entity in the question is uniquely identifiable (e.g. no two families share the same combination of name and address) and if there are multiple possible answers, we check against all of them for determining correctness.

**Sub-tasks for JSON Families**

| Question Type | Number of entities | Question Details |
|---|---|---|
| Family Size | Single Family | Finding the size of a given family in the context |
| Family Member Hobby | Single Family | Checking if a given hobby is a hobby of a specific member from a specific family |
| Family Size Comparison | Multi Family | Comparing size of two given families |
| Family Member Age Comparison | Multi Family | Comparing age of two members from two given families |
| Family Member Hobby Comparison | Multi Family | Comparing hobbies of two members from two given families |

Table 1: Family Question Types

## B.6 ENTITY SOURCES

In addition to constructing the problem, we also provide a list of pre-defined entities to populate the problem with. We prepare train-val-test splits on entity names. For names, we use a dataset of 258k popular baby-names[3]. We use this for both Social Network and Comparison-Age.

For comparing objects, we collected a list of 540 commonly encountered physical objects like "Firetruck" and "Shoebox." We used Gemini-Pro to generate many such candidate objects with mass and size that is consistent and easy to measure. We then asked Gemini-Pro to estimate their size and weight to allow us to construct Congruent statements based on their estimated physical properties.

For incongruent samples we simply reverse the congruent ordering.

We also use a random-string configuration for comparison to test the effect of prior knowledge on congruent and incongruent relationships. Specifically we randomly generated alphanumeric ids of length 5 to replace the name of each object in the comparison. This controls for the effect of prior knowledge as entity names like 3Am4O and gj1Bx have no semantic priors in the language model.

For Syllogism we prompted an LLM to generate a diverse set of plural nouns. The entities are randomly assigned when generating the syllogisms.

For JSON Families, the entities include family names (first, last names), address and details of every member. These are randomly chosen from a set of predefined names, city names, states, postal codes and hobbies. These lists were generated by prompting an LLM to provide a commonly used entities, following which a random combination is chosen when identifying a family / a member.

## B.7 MISC DETAILS

**Heuristic Rejection Sampling**    Because ReCogLab is a data generating process, we do not have precise control of the posterior distribution of examples. This issue becomes prevalent if the goal is to slice the posterior distribution to understand the behavior on a specific subset of generated problems.

One example where we used heuristic rejection sampling is in symbolic distance experiments. Because ReCogLab generates the graph problem first and then a testable question, some kinds of testable questions are impossible to generate given a specific graph problem. To make this concrete, we tested the symbolic distance effect on a random tree configuration of social network. However once the configuration of the random tree has been sampled, randomly sampling two entity nodes to construct a question will tend to produce low symbolic distance questions. This means that high symbolic distances are hard to find in this data generating process. Another example of a rebalancing we use is to generate comparable numbers of determinate and indeterminate examples since in large graphs, indeterminate examples are easier to sample than determinate ones.

---

[3]Source: https://www.cs.princeton.edu/introcs/data/names.csv

To prevent oversampling, we use a heuristic rejection sampling to discard generated examples that have a common property and increase the proportion of rare-property questions. In particular, we annotate metadata like symbolic distance or the answer associated with each example and record a histogram of annotated metadata over examples in the generated dataset. If a newly created example has a metadata that is twice as common as the least common metadata above a certain minimum threshold, we discard that example. Because generating evaluation examples is essentially free, we can simply resample an example until it has metadata properties that we care about.

We choose this approach of balancing test examples since it doesn't require a specific implementation, as long as a process has a probability of generating an interesting configuration we can upsample it's occurrence in the dataset. Additionally some notions of parameters share common meaning but have wildly different implementation strategies. For instance symbolic distance is trivial to calculate for a linear chain configuration, but impossible to control in a random tree. Rejection sampling resolves this issue by allowing us to upsample rare examples without needing to explicitly define how those rare labels generate.

**Flavor Text**   Flavor text is text that adds depth or background to a character or relationship. This embellishment of factual statements adds an additional layer of cognitive load. Instead of simply presenting statements as facts, flavor text rewrites them to a statement that implies two entities' relationship with each other. We generate a total of 83 flavor texts using Gemini-Pro and specific instructions.

## C   ADDITIONAL EXPERIMENTS

### C.1   SYMBOLIC DISTANCE EFFECT

Symbolic distance refers to the number of relational reasoning steps that join two entities in a sequential reasoning task (e.g., if $A > B > C > D$, the symbolic distance between A and D is 3). The symbolic distance *effect* refers to the observation that more distant comparison judgments are, perhaps surprisingly, easier for humans (except at distance 1, where observed pairs can be memorized)

Here we provide further experimental results from Fig. 4 on other tasks. In Fig. 10 and Fig. 11 we show all 4 tasks and the frontier performance of Symbolic Distance and Graph Size. We can see that there's a ordinality to difficulty on these tasks, as reflected in the main paper flavor text results in harder problems and degrades performance (despite being the same underlying grpah problem). This indicates that relational reasoning is not robust against textual distractions.

The one exception to this trend is GPT-4o which shows consistent performance across the 4 various tasks according to the symbolic distance and graph size. This implies a strong robustness to textual distractions and also the ability to perform equally well between variations of the same task. There's also consistent performance between ExactPath and NumHops tasks which indicates a robustness to the format of the question. ExactPath is a version of NumHops that requires the exact order of the answer.

We also see that the performance of each model is dictated by the symbolic distance first, and then the number of entities in the graph. This is indicated by all models have strong performance among problems with symbolic distance $< 5$ and even when the graph size is greater than 40 entities. However a smaller graph of only 20 entities does result in being able to solve higher symbolic distance questions.

### C.2   META-REASONING FPR ANALYSIS

The corresponding figure of results is Fig. 9. For ease of referencing we have reproduced the figure in Fig. 12.

#### C.2.1   DETECTING STATEMENT CONSISTENCY

As we mentioned before, the diagonal elements of the confusion matrix represent correctly classified examples. An additional interesting statistic to analyze is the type of classification errors each model makes. This is represented by the off-diagonal elements and is related a classifiers' false positive

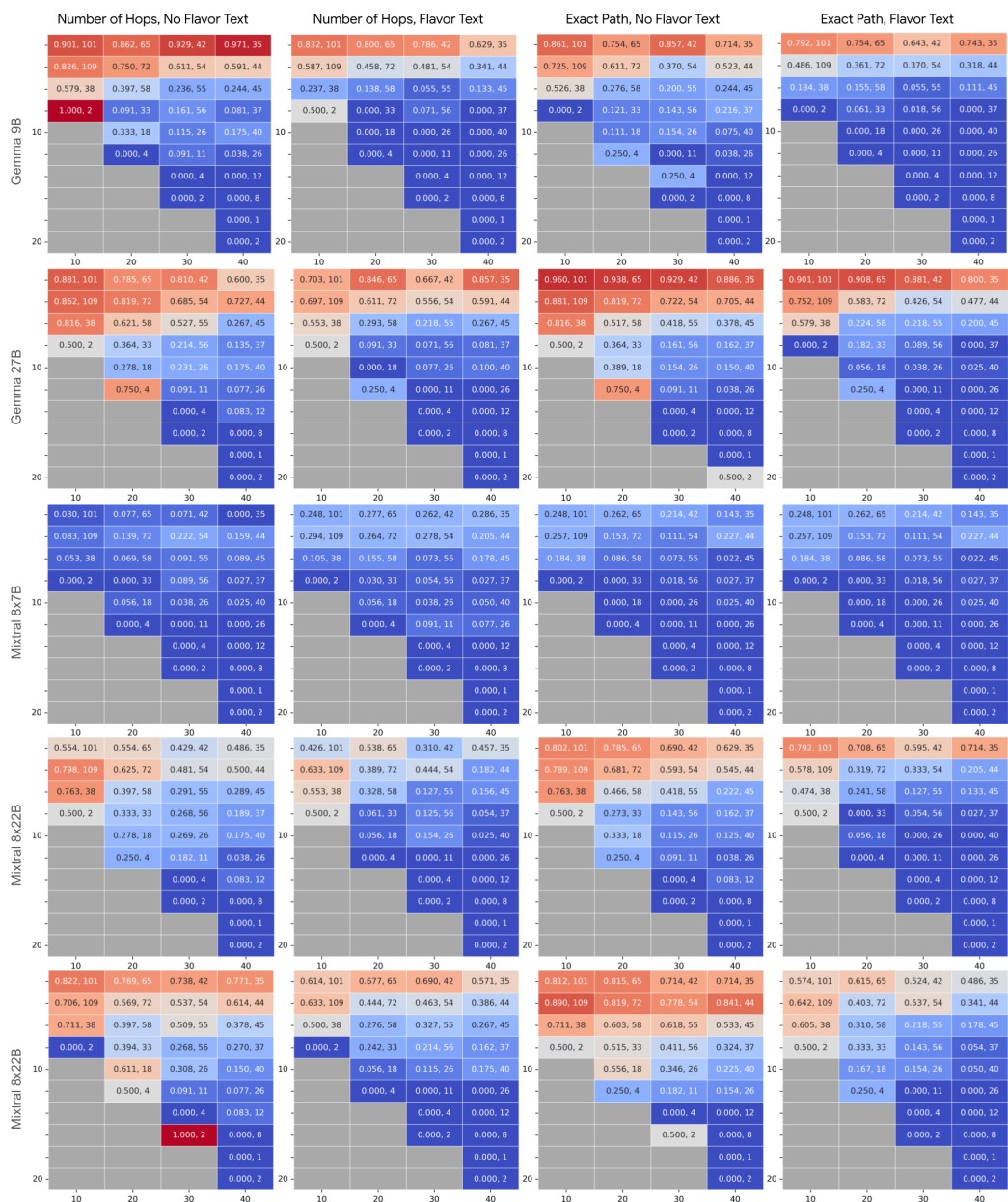

Figure 10: **Symbolic Distance and Graph Size**. Here we show the each model's performance on 4 Social Network tasks. The y-axis denotes performance on examples sliced by the Symbolic Distance required to answer while the x-axis denotes the number of entities in the problem. Each box reports the accuracy and number of test examples in the test set in the slice. Red indicate better performance on a slice.

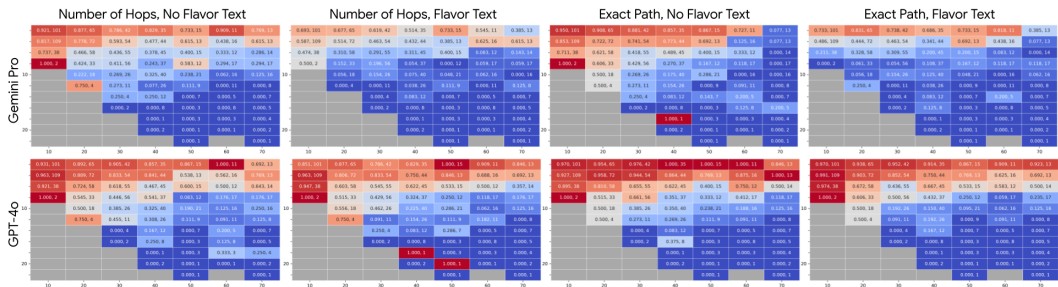

Figure 11: **Symbolic Distance and Graph Size**. Here we show the each model's performance on 4 Social Network tasks. The y-axis denotes performance on examples sliced by the Symbolic Distance required to answer while the x-axis denotes the number of entities in the problem. Each box reports the accuracy and number of test examples in the test set in the slice. Red indicate better performance on a slice.

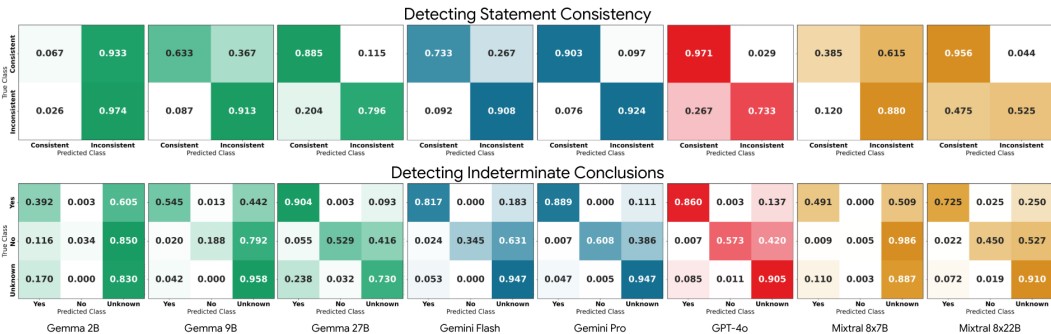

Figure 12: **Confusion matrix on meta-reasoning**. Reproduced copy of Fig. 9

rates (FPR). As an example, the Gemma-2B has a false positive rate of 93.3% when given consistent set of statements.

Because our probes are zero-shot tasks, the models' predictions are inherently uncalibrated. Additionally the decoded predictions are complicated joint distribution of token likelihoods that make it impossible to estimate the logits of a specific parsed prediction so we cannot use ranking metrics. This presents an opportunity to investigate each models' implicit reasoning bias. For instance, the GPT-4o predictions have a higher FPR for Inconsistent premises than consistent (26.7% vs 2.9%), implying that GPT-4o exhibits a preference for answering consistent. Gemini Flash predictions have the opposite bias, with higher rates of False Positive for Consistent premises that indicate a preference to predict inconsistent. Gemini Pro performs the best at zero-shot statement consistency detection and has relatively balanced false positive and false positive rates indicating the best predictive performance.

### C.2.2 DETECTING INDETERMINATE CONCLUSIONS

We show results of this probe in the second row of confusion matrices in Fig. 12. Similar to the previous probe, the diagonals are correctly classified examples and the off-diagonal represent misclassified false positives.

For any determinate example we construct, determining if $A > B$ is as difficult as determining if $B > A$. However we see that there's a significant performance gap across all language models in correctly identifying determinate examples. In other words, presenting the same problem with answer *Yes*, but flipping the question such that the answer is now *No*, will cause the language model to prefer predicting indeterminacy.

One possible interpretation is that language models prefer to answer *Yes*. This is further supported when examining the language models bias towards answering *Yes* on indeterminate examples that are incorrectly reasoned to be determinate. Of note, a similar "acquiescence bias" toward answering *Yes* has been noted in humans in a variety of settings (Krosnick, 1999).

The immediate findings suggest that a self-consistency scheme could improve logical deductions and reduce the implicit reasoning bias towards affirmative answers. Longer term, pursuing strategies beyond instruction tuning to improve detection of indeterminate logical premises is important for safety and human interactions.

# D  ADDITIONAL TEST EXAMPLES

Below we show additional examples generated from our framework for various tasks/sub-tasks/configurations.

## D.1  COMPARISON

### D.1.1  COMPARISON - OBJECTS

**Question:**
Straightener is smaller than Sugar bowl
Paperweight is larger than Oyster
Folder is smaller than Gemstones
Shaving cream is larger than Sand dollar
Eye pin is smaller than Flower arrangement
Sand dollar is larger than Pliers
Oyster is larger than Lego
Sugar bowl is smaller than Watermelon
Drum is larger than Cello
Folder is larger than Flower arrangement
Pill is smaller than Pliers
Ceiling is smaller than Cello
Paperweight is smaller than Pill
Gemstones is smaller than Keyboard
Ceiling is larger than Audio interface
Straightener is larger than Shaving cream
Keyboard is smaller than Lego
Apple is smaller than Audio interface
Drum is smaller than Eye pin
Is Pill smaller than Oyster?

**Answer(s):**
No

**Question:**
Lego is larger than Leaf
Thermostat is larger than Stage
Thermostat is smaller than Trowel
Ribbon is smaller than Saxophone
Sculpture is larger than Saxophone
Cane is smaller than Coffee mug Playbill is smaller than Rake
Rake is smaller than Ribbon
Leaf is larger than Lamp
Banjo is larger than Bandage
Paperweight is smaller than Playbill
Gemstones is larger than Coffee mug
Stage is larger than Sinker
Banjo is smaller than Cane
Bandage is larger than Accordion
Trowel is smaller than Water bottle
Gemstones is smaller than Lamp
Sculpture is smaller than Sinker
Paperweight is larger than Lego
Is Ribbon smaller than Paperweight?

**Answer(s):**
No

**Question:**
Trowel is smaller than Wire cutters
Birthday cake is smaller than CD
Mushroom is larger than Measuring cup
CD is smaller than Cap
Thesaurus is larger than Soda can
Measuring cup is larger than Lego
Leaf is larger than Guitar
Mushroom is smaller than Power strip
Extension cord is larger than Envelope
Wire cutters is smaller than Wood
Conditioner is smaller than Cup
Extension cord is smaller than Guitar
Power strip is smaller than Soda can
Envelope is larger than Cup
Trowel is larger than Tripod
Ashtray is smaller than Birthday cake
Leaf is smaller than Lego
Tripod is larger than Thesaurus
Conditioner is larger than Cap
Is Trowel larger than Birthday cake?
**Answer(s):**
Yes

D.1.2   COMPARISON - PEOPLE

**Question:**
Silver is younger than Trace
Madyson is older than Lorelai
Evertt is younger than Hilma
Rush is older than Petra
Eulah is younger than Eulalie
Petra is older than Orlena
Lorelai is older than Leta
Trace is younger than Vernetta
Cooper is older than Chrissie
Evertt is older than Eulalie
Nigel is younger than Orlena
Ceil is younger than Chrissie
Madyson is younger than Nigel
Hilma is younger than Khalid
Ceil is older than Betty
Silver is older than Rush
Khalid is younger than Leta
Allison is younger than Betty
Cooper is younger than Eulah
Is Nigel younger than Lorelai?

**Answer(s):**
No

**Question:**
Julius is older than Jalissa
Shemar is older than Raquel
Shemar is younger than Treyvon
Lyn is younger than Mae
Natasha is older than Mae

Danyel is younger than Demario
Lonzo is younger than Lori
Lori is younger than Lyn
Jalissa is older than Garey
Cherrie is older than Carmen
Kaleigh is younger than Lonzo
Estefani is older than Demario
Raquel is older than Phoenix
Cherrie is younger than Danyel
Carmen is older than Ayana
Treyvon is younger than Vickey
Estefani is younger than Garey
Natasha is younger than Phoenix
Kaleigh is older than Julius
Question: Is Lyn younger than Kaleigh?
**Answer(s):**
No

**Question:**
Myah is younger than Ozzie
Antonetta is younger than Anya
Hettie is older than Elmore
Anya is younger than Arizona
Marely is older than Marcela
Elmore is older than Donn
Devonta is older than Darryl
Hettie is younger than Iver
Cedrick is older than Case
Ozzie is younger than Thomas
Blake is younger than Briana
Cedrick is younger than Darryl
Iver is younger than Marcela
Case is older than Briana
Myah is older than Mikeal
Abraham is younger than Antonetta
Devonta is younger than Donn
Mikeal is older than Marely
Blake is older than Arizona
Is Myah older than Antonetta?
**Answer(s):**
Yes

D.2 SOCIAL NETWORKS

**Question:**
Arjun is always honest with Marcelo, even when it's hard.
When Arjun needs to talk, Vida is the first one they call.
Casandra is always impressed by Lorena's knowledge and intelligence.
Vida is always willing to listen to Julianne's problems.
Roselyn is like family to Ava.
Exie is always willing to listen to Casandra's problems.
Lukas and Marcelo's families often have dinner together.
Ava is always impressed by Casandra's creativity and artistic talents.
Lorena and Vida enjoy discussing books and movies they've both seen.
Any two friends are able to pass along a message, which allows messages to move from one friend to another. Thus, messages can be passed between two people through friends they have in common.
If Vida wants to get a message to Marcelo as quickly as possible, who should Vida give it to?
**Answer(s):**
Arjun

**Question:**
Austin knows how to make Gunnar laugh, even on a bad day.
Barton and Marlana share inside jokes that only they understand.
Daisey always respects Creola's opinions, even when they disagree.
Branden is always honest with Tierra, even when it's hard.
Tierra can always count on Paola for a shoulder to cry on.
Marlana is always the first person Creola calls with good news.
Marvin and Gunnar have inside jokes that no one else understands.
Branden knows all of Barton's favorite snacks and surprises them with them sometimes.
Marvin knows how to tease Barton without hurting their feelings.
Any two friends are able to pass along a message, which allows messages to move from one friend to another. Thus, messages can be passed between two people through friends they have in common.
If Daisey wants to get a message to Tierra as quickly as possible, who should Daisey give it to?
**Answer(s):**
Creola

**Question:**
Daijah is always there to listen when Elmore needs to vent about work.
Elmore is always willing to listen to Charlee's problems.
You can often find Anwar and Finn laughing and chatting away.
When Walker needs to talk, Alvin is the first one they call.
Charlee knows how to calm Anwar down when they're stressed or anxious.
Minerva always knows how to cheer Elmore up.
Cathi and Anwar have a mutual respect for each other's personal space and boundaries.
Walker and Charlee enjoy trying new hobbies and activities together.
Toma and Cathi have a deep and meaningful connection.
Any two friends are able to pass along a message, which allows messages to move from one friend to another. Thus, messages can be passed between two people through friends they have in common.
If Cathi wants to get a message to Minerva as quickly as possible, who should Cathi give it to?
**Answer(s):**
Anwar

## D.3 JSON FAMILIES

### D.3.1 JSON FAMILIES - FAMILY SIZE

**Question:**
"Family Name": "Wilson", "Address": "544 Pine St, Palo Alto, CA 95841", "Members": [ "Name": "Ava", "Age": 13, "Hobbies": [ "cycling", "music", "reading" ] , "Name": "Grace", "Age": 71, "Hobbies": [ "running" ] , "Name": "Bob", "Age": 91, "Hobbies": [ "writing", "painting", "gardening", "running", "knitting", "cooking" ] , "Name": "Diego", "Age": 93, "Hobbies": [ "music", "running", "dancing" ] ] "Family Name": "Wilson", "Address": "695 Divisadero St, Daly City, CA 70635", "Members": [ "Name": "Frank", "Age": 45, "Hobbies": [ "knitting", "cooking", "cycling" ] , "Name": "Diego", "Age": 67, "Hobbies": [ "reading", "gardening", "music", "writing", "dancing", "cooking", "traveling" ] , "Name": "Bob", "Age": 89, "Hobbies": [ "music", "knitting", "gardening", "painting", "writing", "dancing" ] , "Name": "Liam", "Age": 96, "Hobbies": [ "knitting", "painting" ] ] "Family Name": "Rodriguez", "Address": "48 Lombard St, San Mateo, CA 35388", "Members": [ "Name": "Jack", "Age": 91, "Hobbies": [ "music", "gardening" ] , "Name": "Alice", "Age": 41, "Hobbies": [ "gardening", "writing", "running", "cycling", "reading", "dancing", "music", "traveling", "painting" ] , "Name": "Frank", "Age": 100, "Hobbies": [ "writing", "gardening", "music" ] ] "Family Name": "Brown", "Address": "959 Market St, San Jose, CA 10946", "Members": [ "Name": "Bob", "Age": 36, "Hobbies": [ "running", "music", "painting", "reading", "knitting", "writing" ] , "Name": "Bob", "Age": 10, "Hobbies": [ "painting", "traveling", "dancing" ] , "Name": "Kai", "Age": 88, "Hobbies": [ "cooking", "writing" ] ] "Family Name": "Wilson", "Address": "326 Lombard St, Daly City, CA 99979", "Members": [ "Name": "Muhammad", "Age": 63, "Hobbies": [ "cycling", "knitting", "writing" ] , "Name": "Emily", "Age": 36, "Hobbies": [ "writing", "traveling", "knitting", "reading", "running" ] ]
How many members are in the Brown family living on 959 Market St, San Jose, CA 10946? Answer as a single number.

**Answer(s):**
3

### D.3.2 JSON Families - Family Member Hobby

**Question:**
"Family Name": "Wilson", "Address": "544 Pine St, Palo Alto, CA 95841", "Members": [ "Name": "Ava", "Age": 13, "Hobbies": [ "cycling", "music", "reading" ] , "Name": "Grace", "Age": 71, "Hobbies": [ "running" ] , "Name": "Bob", "Age": 91, "Hobbies": [ "writing", "painting", "gardening", "running", "knitting", "cooking" ] , "Name": "Diego", "Age": 93, "Hobbies": [ "music", "running", "dancing" ] ] "Family Name": "Wilson", "Address": "695 Divisadero St, Daly City, CA 70635", "Members": [ "Name": "Frank", "Age": 45, "Hobbies": [ "knitting", "cooking", "cycling" ] , "Name": "Diego", "Age": 67, "Hobbies": [ "reading", "gardening", "music", "writing", "dancing", "cooking", "traveling" ] , "Name": "Bob", "Age": 89, "Hobbies": [ "music", "knitting", "gardening", "painting", "writing", "dancing" ] , "Name": "Liam", "Age": 96, "Hobbies": [ "knitting", "painting" ] ] "Family Name": "Rodriguez", "Address": "48 Lombard St, San Mateo, CA 35388", "Members": [ "Name": "Jack", "Age": 91, "Hobbies": [ "music", "gardening" ] , "Name": "Alice", "Age": 41, "Hobbies": [ "gardening", "writing", "running", "cycling", "reading", "dancing", "music", "traveling", "painting" ] , "Name": "Frank", "Age": 100, "Hobbies": [ "writing", "gardening", "music" ] ] "Family Name": "Brown", "Address": "959 Market St, San Jose, CA 10946", "Members": [ "Name": "Bob", "Age": 36, "Hobbies": [ "running", "music", "painting", "reading", "knitting", "writing" ] , "Name": "Bob", "Age": 10, "Hobbies": [ "painting", "traveling", "dancing" ] , "Name": "Kai", "Age": 88, "Hobbies": [ "cooking", "writing" ] ] "Family Name": "Wilson", "Address": "326 Lombard St, Daly City, CA 99979", "Members": [ "Name": "Muhammad", "Age": 63, "Hobbies": [ "cycling", "knitting", "writing" ] , "Name": "Emily", "Age": 36, "Hobbies": [ "writing", "traveling", "knitting", "reading", "running" ] ]
Is writing a hobby of Jack from the Rodriguez family living on 48 Lombard St, San Mateo, CA 35388? Answer with Yes or No. Answers: **Answer(s):**
No

### D.3.3 JSON Families - Family Size Comparison

**Question:**
"Family Name": "Wilson", "Address": "544 Pine St, Palo Alto, CA 95841", "Members": [ "Name": "Ava", "Age": 13, "Hobbies": [ "cycling", "music", "reading" ] , "Name": "Grace", "Age": 71, "Hobbies": [ "running" ] , "Name": "Bob", "Age": 91, "Hobbies": [ "writing", "painting", "gardening", "running", "knitting", "cooking" ] , "Name": "Diego", "Age": 93, "Hobbies": [ "music", "running", "dancing" ] ] "Family Name": "Wilson", "Address": "695 Divisadero St, Daly City, CA 70635", "Members": [ "Name": "Frank", "Age": 45, "Hobbies": [ "knitting", "cooking", "cycling" ] , "Name": "Diego", "Age": 67, "Hobbies": [ "reading", "gardening", "music", "writing", "dancing", "cooking", "traveling" ] , "Name": "Bob", "Age": 89, "Hobbies": [ "music", "knitting", "gardening", "painting", "writing", "dancing" ] , "Name": "Liam", "Age": 96, "Hobbies": [ "knitting", "painting" ] ] "Family Name": "Rodriguez", "Address": "48 Lombard St, San Mateo, CA 35388", "Members": [ "Name": "Jack", "Age": 91, "Hobbies": [ "music", "gardening" ] , "Name": "Alice", "Age": 41, "Hobbies": [ "gardening", "writing", "running", "cycling", "reading", "dancing", "music", "traveling", "painting" ] , "Name": "Frank", "Age": 100, "Hobbies": [ "writing", "gardening", "music" ] ] "Family Name": "Brown", "Address": "959 Market St, San Jose, CA 10946", "Members": [ "Name": "Bob", "Age": 36, "Hobbies": [ "running", "music", "painting", "reading", "knitting", "writing" ] , "Name": "Bob", "Age": 10, "Hobbies": [ "painting", "traveling", "dancing" ] , "Name": "Kai", "Age": 88, "Hobbies": [ "cooking", "writing" ] ] "Family Name": "Wilson", "Address": "326 Lombard St, Daly City, CA 99979", "Members": [ "Name": "Muhammad", "Age": 63, "Hobbies": [ "cycling", "knitting", "writing" ] , "Name": "Emily", "Age": 36, "Hobbies": [ "writing", "traveling", "knitting", "reading", "running" ] ]
Which family is larger, the Wilson family living on 326 Lombard St, Daly City, CA 99979 or the Brown family living on 959 Market St, San Jose, CA 10946? Answer with the family name of the larger family. **Answer(s):**
Brown

### D.3.4  JSON FAMILIES - FAMILY MEMBER AGE COMPARISON

**Question:**
"Family Name": "Wilson", "Address": "544 Pine St, Palo Alto, CA 95841", "Members": [ "Name": "Ava", "Age": 13, "Hobbies": [ "cycling", "music", "reading" ] , "Name": "Grace", "Age": 71, "Hobbies": [ "running" ] , "Name": "Bob", "Age": 91, "Hobbies": [ "writing", "painting", "gardening", "running", "knitting", "cooking" ] , "Name": "Diego", "Age": 93, "Hobbies": [ "music", "running", "dancing" ] ] "Family Name": "Wilson", "Address": "695 Divisadero St, Daly City, CA 70635", "Members": [ "Name": "Frank", "Age": 45, "Hobbies": [ "knitting", "cooking", "cycling" ] , "Name": "Diego", "Age": 67, "Hobbies": [ "reading", "gardening", "music", "writing", "dancing", "cooking", "traveling" ] , "Name": "Bob", "Age": 89, "Hobbies": [ "music", "knitting", "gardening", "painting", "writing", "dancing" ] , "Name": "Liam", "Age": 96, "Hobbies": [ "knitting", "painting" ] ] "Family Name": "Rodriguez", "Address": "48 Lombard St, San Mateo, CA 35388", "Members": [ "Name": "Jack", "Age": 91, "Hobbies": [ "music", "gardening" ] , "Name": "Alice", "Age": 41, "Hobbies": [ "gardening", "writing", "running", "cycling", "reading", "dancing", "music", "traveling", "painting" ] , "Name": "Frank", "Age": 100, "Hobbies": [ "writing", "gardening", "music" ] ] "Family Name": "Brown", "Address": "959 Market St, San Jose, CA 10946", "Members": [ "Name": "Bob", "Age": 36, "Hobbies": [ "running", "music", "painting", "reading", "knitting", "writing" ] , "Name": "Bob", "Age": 10, "Hobbies": [ "painting", "traveling", "dancing" ] , "Name": "Kai", "Age": 88, "Hobbies": [ "cooking", "writing" ] ] "Family Name": "Wilson", "Address": "326 Lombard St, Daly City, CA 99979", "Members": [ "Name": "Muhammad", "Age": 63, "Hobbies": [ "cycling", "knitting", "writing" ] , "Name": "Emily", "Age": 36, "Hobbies": [ "writing", "traveling", "knitting", "reading", "running" ] ]
Who is older: Muhammad from the Wilson family living on 326 Lombard St, Daly City, CA 99979 or Jack from the Rodriguez family living on 48 Lombard St, San Mateo, CA 35388? If both are the same age, answer with the name that comes first alphabetically. Answer with the name.

**Answer(s):**
Jack

### D.3.5  JSON FAMILIES - FAMILY MEMBER HOBBY COMPARISON

**Question:**
"Family Name": "Wilson", "Address": "544 Pine St, Palo Alto, CA 95841", "Members": [ "Name": "Ava", "Age": 13, "Hobbies": [ "cycling", "music", "reading" ] , "Name": "Grace", "Age": 71, "Hobbies": [ "running" ] , "Name": "Bob", "Age": 91, "Hobbies": [ "writing", "painting", "gardening", "running", "knitting", "cooking" ] , "Name": "Diego", "Age": 93, "Hobbies": [ "music", "running", "dancing" ] ] "Family Name": "Wilson", "Address": "695 Divisadero St, Daly City, CA 70635", "Members": [ "Name": "Frank", "Age": 45, "Hobbies": [ "knitting", "cooking", "cycling" ] , "Name": "Diego", "Age": 67, "Hobbies": [ "reading", "gardening", "music", "writing", "dancing", "cooking", "traveling" ] , "Name": "Bob", "Age": 89, "Hobbies": [ "music", "knitting", "gardening", "painting", "writing", "dancing" ] , "Name": "Liam", "Age": 96, "Hobbies": [ "knitting", "painting" ] ] "Family Name": "Rodriguez", "Address": "48 Lombard St, San Mateo, CA 35388", "Members": [ "Name": "Jack", "Age": 91, "Hobbies": [ "music", "gardening" ] , "Name": "Alice", "Age": 41, "Hobbies": [ "gardening", "writing", "running", "cycling", "reading", "dancing", "music", "traveling", "painting" ] , "Name": "Frank", "Age": 100, "Hobbies": [ "writing", "gardening", "music" ] ] "Family Name": "Brown", "Address": "959 Market St, San Jose, CA 10946", "Members": [ "Name": "Bob", "Age": 36, "Hobbies": [ "running", "music", "painting", "reading", "knitting", "writing" ] , "Name": "Bob", "Age": 10, "Hobbies": [ "painting", "traveling", "dancing" ] , "Name": "Kai", "Age": 88, "Hobbies": [ "cooking", "writing" ] ] "Family Name": "Wilson", "Address": "326 Lombard St, Daly City, CA 99979", "Members": [ "Name": "Muhammad", "Age": 63, "Hobbies": [ "cycling", "knitting", "writing" ] , "Name": "Emily", "Age": 36, "Hobbies": [ "writing", "traveling", "knitting", "reading", "running" ] ]
What hobbies do Muhammad from the Wilson family living on 326 Lombard St, Daly City, CA 99979 and Jack from the Rodriguez family living on 48 Lombard St, San Mateo, CA 35388 share? List the hobbies in alphabetical order, separated by commas, or answer N/A if they share no hobbies.
**Answer(s):**
N/A

### D.4 SYLLOGISMS

**Question:**
All accounts are actions
No actions are actors
Which of the following is true?
All accounts are actors
No accounts are actors
Some accounts are actors
Some accounts are not actors
**Answer(s):**
No accounts are actors

**Question:**
No accounts are actions
All actors are accounts
Which of the following is true?
All actors are actions
No actors are actions
Some actors are actions
Some actors are not actions
**Answer(s):** No actors are actions

**Question:**
Some accounts are actions
No actors are accounts
Which of the following is true?
All actions are actors
No actions are actors
Some actions are actors
**Answer(s):**
Some actions are not actors

