# OpenReview forum: "ReCogLab: a framework testing relational reasoning & cognitive hypotheses on LLMs"
_ICLR.cc/2025/Conference — ICLR 2025 Poster_

### Official Review · Reviewer_P3cs · 2024-10-21

**Soundness:** 3
**Presentation:** 2
**Contribution:** 3
**Rating:** 6
**Confidence:** 4

**Summary:**

This paper introduces ReCogLab, a sandbox evaluation paradigm for studying memory and reasoning in Large Language Models, drawing inspiration from the cognitive sciences. ReCogLab currently contains a toolkit for building memory and reasoning tasks based on four domains: property comparisons, social networks, hierarchical reasoning (which are called JSON families), and syllogistic reasoning. Each of these domains are formalised as tasks that probe graph-based reasoning. Problems are constructed as (directed) graphs, where each vertex is a person or an object and each edge is some directed relation between those objects. Graphs are built implicitly in ReCogLab and then described only in natural language vignettes. The LLM is then tasked with answering questions about the graphs, also in natural language. For instance, in the Social Networks domain, a graph is implicitly constructed that relates certain members of a social group. The graph is described with sentences such as 'X is friends with Y' and 'Y is friends with Z'. The LLM is then tasked with, for example, describing how a message would be most quickly communicated between two members of the social network, essentially by traversing the edges of the graph. The authors create a testbed of tasks from their four domains drawing inspiration from real experiments done with humans in the cognitive sciences, and study the reasoning and memory capabilities of six large language models. This enables them to study, in LLMs, a number of psychological tendencies that have been observed in humans. Ultimately, they observe many human-like effects emerging in LLMs.

**Strengths:**

This paper has several strengths. First, the production of a sandbox testbed, rather than a static, fixed benchmark, is exactly where evaluation should be going in AI and machine learning. ReCogLab is a toolbox for creating nuanced cognitive experiments, rather than a dataset that someone can take off the shelf and apply without much thought. Moreover, the toolbox significantly reduces the likelihood of future LLMs being trained directly on testing data - contamination is something that has significantly hindered the validity of many new and old benchmarks in the literature. Second, the authors identify a series of psychological effects that few have studied directly in LLMs, but which are key markers of behavioural similarity between us and them, allowing us to diagnose the human-likeness of these systems. Finally, the analyses are comprehensive and balanced. The authors are cautious not to prematurely ascribe human like cognitive capabilities to these systems, noting that there is more work to be done to show this. This is a refreshing contrast to much of the overzealous AI hype from other (non-cognitive-science) researchers in the field.

**Weaknesses:**

There are a number of weaknesses in the paper. First, the authors only evaluate 6 LLMs. The addition of at least some members of the Llama series would lead to the results being more comprehensive of the state-of-the-art. Second, the authors could better review other calls to action for doing cognitive science experiments on artificial intelligence/large language models, as well as other frameworks that offer sandbox evaluation toolkits. I have included a number of relevant citations below. Third, the authors could have played more with chain-of-thought prompting to determine whether this leads to better performance on their tests. Finally, there are some minor linguistic disfluencies. The title is difficult to parse. "A framework testing relational reasoning, cognitive hypotheses on LLMs" contains two disjoint clauses that aren't related. This is off-putting to the reader. Furthermore, the abstract needs to be checked for sense. 'Phenomenon' should be plural 'phenomena' and the sentence started 'while some of these have been studied individually...' needs to be reworded as it doesn't make grammatical sense. Lastly, the term 'indeterminancy' should be replaced with 'indeterminacy' in the latter part of the paper.

References

Hernández-Orallo, J. (2017). The measure of all minds: evaluating natural and artificial intelligence. Cambridge University Press.
Kiela, D., Bartolo, M., Nie, Y., Kaushik, D., Geiger, A., Wu, Z., ... & Williams, A. (2021). Dynabench: Rethinking benchmarking in NLP. arXiv preprint arXiv:2104.14337.
Thrush, T., Tirumala, K., Gupta, A., Bartolo, M., Rodriguez, P., Kane, T., ... & Kiela, D. (2022). Dynatask: A framework for creating dynamic AI benchmark tasks. arXiv preprint arXiv:2204.01906.
Voudouris, K., ..., Cheke, L. G. (2024) The Animal-AI Environment: A Virtual Laboratory For Comparative Cognition and Artificial Intelligence Research. arXiv preprint arXiv:2312.11414.

**Questions:**

1. What is the goal of studying this type of reasoning in LLMs?
2. How might we use the outputs of ReCogLab to better direct LLM development and deployment?
3. Did you run any experiments with Chain-of-Thought prompting? It would be interesting to determine whether this had any impact on performance.

---

> ### Author Response · Authors · 2024-11-22
>
> Thank you for your helpful review and your praise for our motivation. We thought that your description of our work exactly captured what we were trying to accomplish:
> “First, the production of a sandbox testbed, rather than a static, fixed benchmark, is exactly where evaluation should be going in AI and machine learning. ReCogLab is a toolbox for creating nuanced cognitive experiments, rather than a dataset that someone can take off the shelf and apply without much thought.
>
> We will now address your questions and comments.
>
> Number of LLMs
>
> Due to legal issues with the LLAMA license, we cannot add this specific comparison. However, we did add Mixtral 8x7B and Mixtral 8x22B open source models to our comparisons, bringing the total language models we evaluate to eight (5 open, 3 proprietary). We hope this helps alleviate this concern.
>
> Additional sandbox evaluation toolkits
>
> Thank you for the citations. We have added these references to our related work.
>
> Chain of thought
>
> We actually did play with chain-of-thought prompting in our experiments when we chose prompts through cross validation. We apologize if this wasn’t clear in our original draft, please see Appendix B for more details. Some cursory analysis on the validation set performance shows that chain-of-thought outperformed direct prompting. We also find that role-prompting was relatively ineffective.
>
> Typos/Grammar etc
>
> Thank you for the corrections and suggestions. We will change the title in the pdf to be more clear and have made the other fixes you suggest as well.
>
> Why study this type of reasoning
>
> P3cs: What is the goal of studying this type of reasoning in LLMs?
>
> This type of reasoning is of interest because it is hypothesized to be fundamental to the process by which we convert disconnected experiences into a coherent body of knowledge: it involves “stitching together” disparate episodes in such a way that can support inferences about un-observed relationships (see Behrens et al 2018 for more discussion on this).
>
> In psychology and neuroscience, this is important for understanding how experiences contribute to learned knowledge, how that knowledge is represented, and how it is used to support reasoning. For LLMs, it is relevant to understanding how LLMs can similarly integrate information accurately across different locally related passages to make global insights. It also relates to multihop reasoning abilities, continual learning (relating novel associations to previous observed associations),  and capacity limitations on reasoning in general.
>
> Besides the motivation of finding possible similarities between human and LLM reasoning, studying this type of reasoning can help us improve language models. See below.
>
>
> How we might use this for development
>
> We believe that there are several interesting ways this framework can contribute to developing reasoning capabilities in language models.
>
> Because we are able to control the generation of eval benchmarks, we can precisely identify model capability breakpoints. (e.g. X’s relational reasoning capabilities degrade to chance when there are more than 30 entities, Y does not display emergent relational reasoning capabilities over Z because we observed capabilities in Z on similar but easier relational reasoning tasks)
> Instruction-tuning is a standard-way of finetuning a pretrained language model. Our generative framework contains all the necessary parts to instruction tune (Question and Answer in human readable format).
>
> Debiasing Language Models for Fairness. Isolating and understanding biases is important for AI Safety. We’ve already outlined some of the cognitive biases, but ReCogLab makes it easy to probe for social biases in language model performance. One could construct social networks with only feminine names and social networks with only masculine names trivially to understand whether gender has an effect on reasoning capabilities.
>
> AI Safety: Some of our experiments on detecting logical inconsistencies or indeterminate conclusions represent important safeguard reasoning capabilities and our framework allows for us to generate evaluation examples that target these reasoning capabilities specifically (and in many different configurations). For example, if a model is able to correctly identify when something is inconsistent or that there’s insufficient information to draw specific conclusions, it can take more informed action that minimizes risks to humans.

---

> > ### Comment · Reviewer_P3cs · 2024-11-25
> >
> > Thank you for your response to my questions. As I suggested in my original comment, I am generally in favour of projects such as this and think that ReCogLab is a good contribution. I am pleased to see that two further models have been studied.
> >
> > Using precise evaluations to define the operating range or capability frontier of a model, allowing us to administer more useful training examples, is indeed a major advantage of these kinds of evaluations. I think these benefits can be drawn out in the paper, and connected to the relevant literature. Lev Vygotsky's notion of the Zone of Proximal Development (ZPD) seems relevant, and in fact, this is used to construct better training curricula for reinforcement learning agents in the deep mind paper cited below (Adaptive Agents Team, 2023).
> >
> >
> > Team, A. A., Bauer, J., Baumli, K., Baveja, S., Behbahani, F., Bhoopchand, A., ... & Zhang, L. (2023). Human-timescale adaptation in an open-ended task space. arXiv preprint arXiv:2301.07608.

---

> > > ### Author Response · Authors · 2024-11-25
> > >
> > > We're glad that you find our contribution valuable.
> > >
> > > We're happy to add these additional citations in the final draft.

---

### Official Review · Reviewer_G6qF · 2024-10-27

**Soundness:** 1
**Presentation:** 1
**Contribution:** 1
**Rating:** 3
**Confidence:** 3

**Summary:**

ReCogLab provides a series of benchmark tasks for relational memory. Although the tasks themselves do not necessarily seem problematic, the coherence of this series of benchmarks was unclear. Why were these tasks used as opposed to others? The paper would be greatly improved explicitly stating the criteria for selecting these tasks and articulating how these tasks relate to different theoretically distinct aspects of relational reasoning. Building specific hypotheses with respect to different cognitive operations may allow the reader to better understand how to position or understand differences between tasks, especially with the potential for different results. As part of this. the authors could more explicitly articulate specific reasons LLMs (either in general, or specific LLMs) would perform on some tasks compared to others.

In the background, the authors note that some of these tasks have been used in isolation, but there was limited information about how LLMs did on those tasks, and other than having all together in one set, how these benchmarks were preferred to the ones in the literature. Please provide summaries of previous LLM performances on related tasks. What do the authors see concretely about using this set as opposed to the previous tasks?

If this is a benchmark, is it necessary to have human data? Do humans perform differently on these tasks, and might that be a reason that the different benchmarks are needed? The authors should consider whether human data would help understand LLM performances on these tasks, or whether human data would provide insights into how to understand differences between the tasks. What results would the authors expect?

It is unclear why this is relational memory as opposed to relational thinking. What aspects of this make the processes specifically memory related? Is memory necessary for solving these tasks in a way that is different than other LLM tasks (one could make the argument that all LLM tasks require some degree of memory).

In sum, the weaknesses of this paper come from inadequate framing and positioning of its importance in the larger literature. It is possible with more information that the strengths of the paper would be more clearly understood.

**Strengths:**

Examining the capacities of LLMs is important, and it is important to understand the more difficult aspects of potential cognition such as relational reasoning. Establishing benchmarks is an important part of building this research program.

**Weaknesses:**

Although benchmarks are important, it wasn't clear from the writing why these benchmarks are the ones to be used. A disucssion about previous benchmarks is needed with attention to their limitations. The authors main claim is that there isn't a set of all these benchmarks together, but one should simply stitch together other benchmarks as a set. A discussion of why this is not a good strategy is needed. If these are benchmarks, human data may be needed to understand what good performance is.

**Questions:**

1. Why are previous benchmarks inadequate?
2. Why are these methods preferred?
3. Why do LLMs succeed or fail on these different benchmarks?

---

> ### Author Response · Authors · 2024-11-22
>
> Thank you for your useful feedback and validation of the importance of benchmarking in this area of reasoning. We will respond to you comments and questions here.
>
> How is this paper different from literature?
>
> Please see the general response where we go into detail about the motivation of our work. To answer the question directly, our work is different from the literature in that we provide not just a single dataset to study a particular cognitive effect, but a generative framework for flexibly generating datasets to isolate particular effects. Our experiments show that our framework is able to generate a wide variety of different experiments, including some effects such as the symbolic distance effect or the ordering effect that have been noted in both Cognitive Science and LLM literature. No other datasets provide this kind of flexibility.
>
> Relational thinking
>
> “It is unclear why this is relational memory as opposed to relational thinking. What aspects of this make the processes specifically memory related? Is memory necessary for solving these tasks in a way that is different from other LLM tasks (one could make the argument that all LLM tasks require some degree of memory).”
>
> This is a great question, and we will include some text in the discussion on this point. These processes probe memory in that they involve recalling information from the past (in this case, recalling information from context). In this sense, all LLM tasks involve memory, although they don’t necessarily *challenge* memory in the same way as e.g. needle-in-a-haystack, where recalling the required information is made specifically more challenging by embedding it in a large context.
>
> Relational memory refers specifically to memory for relations – recalling that A relates to B and B relates to C. Relational reasoning refers to using this information to reason over these relationally structured memories. What enables this dataset to probe relational memory is that it engages specifically remembering and reasoning about relations, and what makes this a relational memory study is that we are manipulating variables that affect memory: the complexity of the relational structure that is stored in memory and the reasoning process that occurs on top of that.
>
> Relational thinking is a term that tends to be used more in the developmental psychology literature to describe to concept formation in children [e.g. Alexander et al 2016, Starr et al 2023]. It has some conceptual overlap with relational reasoning and memory but is generally used in a different set of studies than the ones we are referencing – we can include a note about this in the discussion [Alexander et al 2016, Starr et al 2023].
>
> Why (these specific) tasks
>
> Please see our general response for this question.
>
> Previously used tasks
>
> “In the background, the authors note that some of these tasks have been used in isolation, but there was limited information about how LLMs did on those tasks, and other than having all together in one set, how these benchmarks were preferred to the ones in the literature.”
>
> We apologize, this may have not been stated clearly enough, allow us to clarify.
> What we meant is that some of the task ideas were used in other works. We did not mean that we literally used these other datasets and ran the exact same experiments. For instance, our syllogisms task is similar (but not identical) to the one studied in (Eisape et al., 2024).
>
> Please also refer to our general response about why what we are doing is not just concatenating previous benchmarks together and how our work is providing a framework for creating new datasets on the fly.
>
>
> Do we need human data
>
> “If this is a benchmark, is it necessary to have human data?”
>
> Because our motivation is to have a framework that allows for the on-the-fly creation and customization of datasets, necessarily our data has to be procedurally generated from templates. Otherwise, each time we needed to in some way change the parameters of the dataset we would have to collect human data. The main drawback of using templated language as opposed to collecting human examples for each data point is generally that templates will have less variety as it’s a constraint on the forms that sentences can take determined by your number of templates. However, because we are not training the models on these, this lack of variety is not as big an issue. And this tradeoff is again necessary for the goals of the dataset. We would push back a bit on this idea that this is not “human data.” While it’s templated, at least for the non-JSON tasks (which is human-readable, but obviously not naturally formatted language), these are valid sentences that people would use; we just use some of the variety.

---

> > ### Comment · Reviewer_G6qF · 2024-11-23
> > **Additional questions**
> >
> > Thank you for your responses. I see the importance here of having new dynamically generated benchmarks, but I was wondering if you could spend a little more time talking about the theorectical importance of this particular set of benchmarks. Given that you have multiple tests, is this a single ability? Multiple abilities? It would be helpful to understand the computational dynamics that you are trying to assess to better understand how these particular tests help you get to understand whether LLMs are succeeding or not.
> >
> > I do like the idea of dynamic benchmarks, but do you have concern over scoring? If I develop a new model, and I don't have your exact tests, can I compare my results to yours?

---

> > > ### Author Response · Authors · 2024-11-24
> > >
> > > To answer your last question, yes. We’ll be releasing all the dataset generated for our experiments as well as the code that generated it, so future researchers will be able to compare directly if they wish. And, because of the way we handle seeded generation (See Appendix B for more details) with the configuration files, people can identically regenerate it themselves if they wish.
> > >
> > > For the first question, can you clarify what exactly you’re asking us for? What do you mean by “theoretical importance”? We discuss in the general response some of our motivations for the individual tasks, but the motivation for all of them as we say is to study relational reasoning.
> > >
> > > When you say we have multiple “tests” are you talking about our different tasks or about our different experiments? Each of our tasks is studying relational reasoning, but in different problem setups and in different ways. Our different experiments are testing different effects; I would not say these are different abilities, these are more like studying different questions about relational reasoning
> > >
> > > Another way to think about our framework is like a laboratory. If you think of yourself as a scientist and you’re trying to test a hypothesis about an LLM, what do you want to do? Create an experiment. Say you want to know if ordering matters or about distance effects. Our framework can generate the datasets for that. The goal is to make our framework such that scientists can study a variety of different effects (that fall broadly into relational reasoning) very quickly and easily.

---

> > > > ### Comment · Reviewer_G6qF · 2024-11-25
> > > > **Thanks**
> > > >
> > > > Reading through your most recent version I believe clarifies a lot of my questions and the question posed by other authors. The paper is much improved because of it. One place that I am still struggling, although I admit that I can see your point about flexibly generating tasks as important, is that this makes new tests tricky unless one reruns all models on a newly generated set. I am imagining that I build a new model, I fine tune a new model, or a new version of GPT is released. If I am reading this right, the dynamic aspect becomes a liability. This is particularly true with the rate that models are released. I like that one can provide a set of current examples, but then the solution to the dynamic problem is not to use it dynamically. I am sure that you have thought through these issues, and if you can spend some time talking explicitly how you see this being used in the future that may be helpful for the readers

---

> > > > > ### Author Response · Authors · 2024-11-25
> > > > >
> > > > > We’d like to clarify exactly the core of your concern here because I think we don’t quite understand it.
> > > > >
> > > > > If your concern is just that the flexibility and ability to generate multiple datasets makes it impossible to compare experiments when new models are released you won’t be able to directly compare, that is not an issue. We are releasing the exact datasets and the framework that allows them to be deterministically regenerated. Any new model can be rerun on our experiments.
> > > > >
> > > > > Is the concern that the dataset leaks and then we would need to regenerate the dataset and rerun that regenerated dataset on all the previous models? If that’s the case we are happy to provide some additional analysis in the final paper to show that changing the entities has little effects on final results. For this concern, we would emphasize however, that all evaluations have this issue of models being specifically trained on the evaluation data. Our ability to easily generate new versions of the dataset is something that non-generated datasets would in fact find impossible. So this, we believe, is an additional strength for our evaluation.
> > > > >
> > > > > Would you be able to clarify which of these is your concern (or neither), and we can try to talk in more detail?

---

> > > > > > ### Comment · Reviewer_G6qF · 2024-11-26
> > > > > >
> > > > > > I apologize if this is not really an issue, but I am just struggling with the idea that the method generates new criteria, but it is also a benchmark. For a benchmark to be a benchmark, I would think that you would want the evaluation metrics to have constant meaning. A 80 today is the same as an 80 tomorrow. With a dynamic dataset, an 80 today is 80+/- some value and then the same with tomorrow, but the +/- is different. I agree that one can release the criteria at any given point, and so the whole evaluation can be redone at any point, but is relational memory so important that the whole set of evaluations needs to be redone that many times? I am perfectly ok with this being the answer, but moving targets seeems like a disadvantage on a non-central metric rather than an advantage. I would just like to see this fully explored.

---

> ### Author Response · Authors · 2024-11-26
>
> Okay, I think I understand the confusion now. I think something you might be assuming here is that the only purpose you might have to create a dataset is to have a static evaluation for some particular *thing* you're trying to measure about models. But that's not the only thing you can do. In our paper we are showing how you can generate new datasets to test different things. Some things we think people can do with out datasets.
> 1. Studying cognitive effects that appear in humans in LLMs. This was one of the motivations for our work, the lack of flexible datasets in the literature to study many cognitive effects. This is shown e.g. in Figure 4&5.
> 2. Debugging where a model struggles or where it fails. For instance, for the capacity experiment in Figure 3: you could use this to diagnose after how many logical steps (e.g. for comparison) a model tends to have degraded performance.
> 3. Using our framework as a static benchmark. While that wasn't our only or primary goal, we would say that the capacity experiments in Figure 3 can be used as benchmark for long-context problems which require reasoning. For this, the meaning here is constant, it is just more than one number (4 tasks) and is in increasing order of difficulty. And because it scales in difficulty with more entities, it will stay a relevant benchmark because the difficulty can scale as models get more and more powerful.
>
> Hopefully this helps answer your question. We really hope you don't knock our work not entirely focusing on having a static single evaluation (although like we said, it can be used that way).
>
> We hope that these clarifications have helped you see the value of our paper and that you accordingly increase your score and recommend acceptance.

---

### Official Review · Reviewer_UYz6 · 2024-10-29

**Soundness:** 2
**Presentation:** 3
**Contribution:** 2
**Rating:** 6
**Confidence:** 4

**Summary:**

This paper does several things:

* It introduces ReCogLab, a way of generating data-sets for testing relational understanding in LLMs
* It evaluates several LLMs on several relational understanding tasks, noting successes and limitations, and speculating on the connection to human reasoning about relations.

**Strengths:**

The paper has several welcome aspects:

- Generally speaking, relational understanding is a basic part of cognition, so the topic itself is important.
- Testing LLMs on their basic understanding of relations is useful for our understanding both of these models and of human cognition.
- The paper includes more than a single example or a single model, allowing us to (in principle) draw more general conclusions
- The 'stress testing' done by increasing the number of variables/nodes/edges is useful for assessment.

**Weaknesses:**

While there are many positive aspects to the work, and I think it is generally in the right direction, there are some specifics to the paper that make it difficult to recommend acceptance:

Some general comments:

* While I understand the authors' decision not to provide full details of their framework, the fact that they present it as a major contribution of the paper while at the same time not providing full details of how the framework works and promising to do so only upon publication makes it difficult to assess it.

* While I appreciate the use of several tasks and models, it isn't clear or justified why *these* tasks are specific to relational understanding (and not others), and the use of closed models makes it difficult to assess what is actually going on beyond saying "these models do it well, those models don't". I get that many of us are currently assessing models in this way and I do think it is useful to assess the state of the art, but at the moment the assessment of closed-models by mega-corporations falls more into doing QA for them than being able to say something scientific about why and how LLMs fail or succeed on these tasks.

* The connections to human reasoning and psychology are welcome and overall ok, but there are many places where they could be improved (see more details below). Also, I found several of the cases of arguing along the lines of "the models are doing poorly and people also do poorly so maybe they're the same" to be speculative at best and not digging into the phenomena, again see more details below.

* There are many cases where statistical claims are made without statistical tests to back them up. While I agree the tests would likely back up the claims I don't think severe of the claims are resting on solid ground at the moment (again, see more details below).

Comments in more detail (these are not in order of importance but more in order of reading the paper, so some of these are truly minor and had no real affect on the final assessment, they are provided in case they help the authors tighten the work):

-- Not terribly important but the first paragraph is making a bunch of claims that would ideally be backed up by references. For example, "While recent work on memory in large language models (LLMs) assumes that memory refers to the recall of a specific piece of information from the past" -- which recent work?, or, when you say "When humans remember events, facts, places, etc., they don’t just recall disconnected pieces, they recall the associations" -- what is backing this? Similar comments apply to much of the second paragraph

-- Given the importance of relational reasoning that the authors emphasize and I agree with, I found it a bit odd that when we do get to references to back it up in humans it is to a 1976 paper on a specific phenomena of symbolic distancing, a 2010 general textbook, and a 2018 neuroscience paper on episodic memory in the hippocampus. This feels like a "grab bag" of disconnected things.

---- Side note, related to the above: With all respect to Domjan (2010) I found the frequent reliance on an undergrad psych text-book throughout the paper odd. It would be good to back up your claims with regards to more specific papers in the literature.

-- As mentioned in the general comments: I'm not knocking the specific things you decided to test too hard (syllogisms, comparisons, etc) but it would be good to provide some stronger justification for why *these* specific things and not others. Do they specifically cover a majority of relational reasoning? Are they uniquely interesting in some way? I'm not saying they don't or aren't, but the reader shouldn't have to do that work. Also, while we're here: Given the weight and importance you attach to identifying inconsistent premises later in the paper, they seem to come out of nowhere because they aren't really justified in the introduction and given weight or airtime in the figures and background and text that discusses the other cases you're looking at.

-- I said I wouldn't knock the specific tests too hard, but I think the 'social networks' one deserves even more justification, given that it doesn't seem to actually be about 'relations' or 'social reasoning' but, as the authors themselves point out later, navigation of graphs independently of the 'relations' between people. It doesn't really seem to be about relational understanding in the way that, say, "X is on Y". The results would presumably replicate with just "X is to the right of Y" or various other room descriptions. So, why go through stuff like "X is the child of Y, Z never shares T's secrets..."; it all seems kind of unnecessary, and would probably not really work in humans. The interesting stuff about enemies and non-relations is mentioned by the authors but not explored further.

-- super-duper-minor: For figure 1 it would be nice to have the colors match up with the text. For example, the setup for comparison is "orange > bottle > tv > dog", yet in the text dog is red and orange is blue, while in the graph the red node is 'bigger' than the blue node and is separated by only 1 node from it.

-- I get the focus on LLMs and it is good to keep it, but it would be nice to at least acknowledge briefly in the background that other models have tried to model this stuff that don't rely on LLMs. The entire field of Inductive Logic Programming seems relevant, for example.

-- The first paragraph in the background is very disjointed, it makes 3 separate points that don't connect.

-- "Substantial work has compared reasoning in LLMs to humans" -- it is odd to follow this sentence up with a reference to Stupple & Ball (2008) or Evans et al. 1983, which took place before 'LLMs' were a term. I'm not sure if the authors were tripped up by the use in the paper of "parallel processing" (which doesn't mean what it means in PDP connectionism).

-- I appreciate mentioning relevant LLM papers in the background section but it would be good if the authors further delineate what their work contributes beyond the already substantial contributions of those works, right now it isn't particularly clear what the contrast is.

-- ״because friendship is a symmetric relationship, social networks are undirected graphs״ -- while it is true that many social networks are modeled to a first approximation using a simple undirected graph, I'm sure people in social psychology would balk at the statement that 'friendship is a symmetric relationship'. Also the consequent doesn't follow, social networks describe friendships as well as rivalries, hierarchies, family relations, and many other things besides not captured by a simple representation along these lines. As mentioned above, it is not clear to what degree this task is about 'relational understanding' more than it is about navigating an undirected graph, where the graph could describe spatial relations, concepts, or nonsense.

-- it wasn't entirely clear to me how you check for grounding in a generalizable way. That is, I understand that in this case specifically, you used a different LLM to generate the sizes and weights of things, but if I wanted to ensure that my data generally matches reality for some arbitrary property, this seems like a currently unsolved problems in LLMs so we can't simply say 'oh, we'll use LLMs to make sure it is grounded'.

-- I appreciate the decoupling of prompting, but what are the actual details of how this decoupling happened? Which prompts did you use and why?

-- "publicly available language models" doesn't mean "open"

-- There are several cases in the paper where the authors make claims that are meant to be backed up with a statistical test.

For example:
" In Fig. 2 we see this experiment for Comparison and Syllogisms. For Comparison, we can clearly see across nearly all models randomizing the order causes noticeable degradation (Gemma 2B and 9B perform near or below chance of 0.5, guessing or giving invalid answers). Reversing the presentation order also often negatively affects performance in many, but not all cases."

'We can clearly see' is not a statistical test. Presumably you want to use an ANOVA here or one of a myriad of other ways of assessing this claim. Similarly 'negatively affects performance in many, but not all cases' is a non-exact, non-specific way of putting a result.

* "For most models we see a curve in which accuracy decreases with symbolic distance for very small symbolic distances, but then improves for symbolic distances >15." -- Again, this is highly inexact. What do you mean 'most models', what do you mean 'improves above 15'? What statistical test are you running to say these statements? Are you comparing a linear fit to a quadratic fit, for example? Are you simply fitting a linear line and looking at the rise? Are you doing a piece-wise and finding a transition point at 15, and what is associated value of the parameters for any statistical test you ran?

* ""Gemini Pro actually levels out rather than decreasing and GPT-4o only decreasing slightly, suggesting that they are particularly good at needle-in-the haystack style questions" -- here and elsewhere, while I appreciate the speculation, we see the problem with evaluating these closed models, we simply have no way of knowing what is going on.

-- The authors should probably face head-on the fact that spatial distancing in these models isn't working like it does in people. In people it drops from a distance of 1 to 2, but then steadily rises. For these models it seems (to the degree that one can say anything) more like a u-shaped function.

-- I didn't understand why you tested the effect of capacity in the ranges you did, sometimes it seems to go 0-100, sometimes 0-50, sometimes 1-20, sometimes 5-30. In cases where we see degradation it is reasonable to argue it exists, but in cases where we don't (JSON family, values 1-20) it isn't clear if this is a result of not going beyond the current range.

-- Super minor hair splitting: 'nonsense relations' like 'Benny blexed Kelly' is not 'inconsistent' with the real world in the same way that 'iron is lighter than air'.

**Questions:**

* It would be good if the authors could provide more details on their framework and how it works.

* It would be good if the authors tested their frameworks on open models in a way that allowed them to draw scientific conclusions about why some models fail and some don't or show the effects they do.

* It would be good if the authors justified their tasks (I'm not saying they aren't good tasks, I'm saying the authors should motivate them further). The introduction and background could use major tightening and also moving up the fact that you tested indeterminate reasoning and inconsistent premises, and also to explain more how this differs from other major works recently looking at relationship reasoning.

* It would be good if the authors connected this better to the specific psychological literature

* It would be good if the authors provided statistical tests to back up their claims.

---

> ### Author Response · Authors · 2024-11-22
>
> Thank you for your very detailed review. We have incorporated your feedback into our draft and we think that it has made the paper better.
>
> We are happy with your assessment that our paper: studies an important part of cognition, tests many examples and models to draw more general conclusions, and increases the number of variables/nodes.
>
> We hope that you take a look at our general response and reiterate that our main goal was to create a framework which can generate datasets for comparing and evaluating language models along many different dimensions. Thank you for your kind words about the strengths and we are glad you found these aspects of our analyses interesting. We do just want to drive home the point that the strength of our work is that we created a framework that allowed for these analysis in the first place.
>
> We will now go through your questions and other feedback, covering in particular your main points for improvement: framework details, why these tasks, improving connections to human reasoning and psychology and statistical claims. Please follow up with us for any additional points of clarification.
>
> Framework details
>
> Our apologies for not providing enough details. We had an Appendix A which went over more of the framework details, but we have expanded it and added more details that will hopefully make it more clear to reviewers how it works. Please let us know if there are any additional details or clarifications on this before the discussion period ends.
>
> Why these specific tasks
>
> Please see our general response where we try to answer this question. We have also added some additional text in the draft to make this clear to future readers as well.
>
> Why Social Networks Specifically (and related concerns)
>
> We touch on this in the general response, but to be more specific about why we added this task:
>
> For the social networks, we drew specific inspiration from Kumaran & Maguire 2005, who use social networks to probe non-spatial relational memory. Other work (e.g. Whittington et al 2020; Behrens et al 2018) also use social connections (specifically family structures) to exemplify “abstract” (ie non-spatial) relational reasoning. It is exactly because social networks are in a sense the paradigmatic example of abstract relational reasoning in psychology that we use it here.
>
> To be clear, other kinds of relationships (“X is on Y” “X is to the left of Y”) are also important and well-studied types of relational reasoning, but relational reasoning is not limited to physical/spatial reasoning [Battaglia et al 2018, Behrens et al 2018, Cohen & Eichenbaum 1995].
>
> We note that the capacity to generate other experiments of this form is doable with this dataset generator, which we see as a major part of the contribution to computer science. However, the goal of this work was not to be comprehensive but rather to provide a dataset generator and illustrate its capacity to generate relational memory experiments.
>
> Regarding the flowery language, we included this condition exactly to see whether increasing complexity of the language, not complexity of the structure, would affect the LLM. This is a knob we can turn easily with our dataset generator. We did include the simpler text (“X is friends with Y”) to adhere more closely to existing work in psychology, and we agree that it is an open question whether the results with the more flowery text would replicate in humans. However, the ability to generate experiments beyond that which has been studied in humans seems more like a strength than a weakness of the dataset generator. Rather, we see it as a contribution to cognitive science / psychology that we are releasing an easy-to-use parametric generator of relationally structured, text-based problems that include features that haven’t been studied.
>
> We will update the text to add context for why we included this study with these references.
>
> Statistical claims / analyses
>
> Apologies that we did not do more here. We have added a 95% CI over our results so that we can draw more stable conclusions about these models. While these CI’s do validly tell us the 95% for each of these specific models across our dataset, we should be careful to point out what this means. For 4o, for instance, we are sampling n times over the randomness of the model and over the randomness over the dataset, but this is, for instance, not the n we might get from psychology where we have n subjects. So these CI’s should be carefully interpreted to mean our confidence over the randomness for one specific model and configuration. In an ideal world, we would train N different language models from scratch and then get a CI or statistical test over those results to see if these hypotheses hold, but this is obviously not practical.
> We have additionally gone through the specific claims in the paper draft and rewritten these to be more careful in exactly what it is we are claiming.

---

> > ### Comment · Reviewer_UYz6 · 2024-11-24
> >
> > Dear authors,
> >
> > Thank you for your detailed response and follow-up!
> >
> > A few minor things in response:
> >
> > * 'flowery language': you seem to be making the assumption that there's an underlying relationship X--Y ('X is friends with Y') which the flowers language merely obscures to some lesser or greater degree but shouldn't actually matter in terms of the underlying link; but my point was that given the test 'how would you get a message fastest to from [X] to [Z]' this so-called flowery language may matter a great deal. I the language is 'when Arjun needs to talk, Vida is the first one they call' as opposed to "Daisey always respects Creola’s opinions even when they disagree" then people might see the Arjun-Vida connection as 'faster' or 'more reliable' to get a message through then Daisy-Creola. Again, this doesn't really matter to the evaluation of the current paper but I do think further experiments with people somewhere down the line on this might prod this assumption.
> >
> > * statistical tests: I greatly appreciate the addition of confidence intervals alongside the cautionary notes about interpretation. However, I'm still left a bit confused about how I should interpret them. What do you mean "n times over the randomness of the model and over the randomness over the dataset"? Do you mean that you are picking some sub-set of the data and running the model over that sub-set, and then repeating this procedure n times?
> >
> > Also, even with the confidence intervals in place, I still don't see the actual statistical analyses to back up the claims that are being made on top of them. Just as a few examples:
> >
> > E1) "In Figure 5, we report the effect of symbolic distance on performance for our models. We generate Comparison examples of 20 entities and then sample comparison questions based on the symbolic distance. For models that perform above chance (all except Gemma 2B and Mixtral 7B), there is a clear U shape.... LLMs similarly show elevated performance for the shortest and longest symbolic distances, "
> >
> > What is the _statistical test_ that backs up this claims? With the benefit of CI's it seems like everything is noisy and muddled, in a way that makes it hard to see a 'clear U shape'. This is not a _problem_ with variance or CI's, this is _useful_ because it shows us that we _can't_ just 'look at the graph' and 'see a U shape', you need to back this up with comparing, say, a linear to a quadratic model here and looking at the relevant parameters. I also don't know that find 'elevated performance for short/long distances' as opposed to middle ones given these CI's, that's why you need a statistical test.
> >
> > E2) "In Fig. 2 we see this experiment for Comparison and Syllogisms. For Comparison, we can clearly see across nearly all models randomizing the order causes noticeable degradation (Gemma 2B, 9B and Mixtral 7B perform near or below chance of 0.5, guessing or giving invalid answers). Reversing the presentation order also often negatively affects performance in many, but not all cases."
> >
> > Reversing the presentation order affects 'Comparison' negatively in 2 out of 8 cases, as far as I can tell by squinting at the CI's, this is not 'many but not all'.
> >
> > * Citations, Domjan: this is super-duper-nitpicky and has no implication for the scoring of this paper but again, relying on a textbook in this way is still kind of weird. Imagine if someone submitted to ICLR a paper about transformer architectures and instead of citing the original papers on this architecture, cited a later 'Introduction to NLP for CS undergraduates' textbook, then defended their choice by saying that 'transformers are well established at this point' and 'other people did it too'.
> >
> > * Concerns about closed models: I'm sorry if my comments felt like I was taking out frustrations on you, this was not the intention or point. However, 'closed' did not just mean 'we don't get to see the weights', it's that we also don't have access to the many of the decisions, training data, and additional bells and whistles that would perhaps allow us to draw conclusions about these models. It isn't clear to me what 'science' we are doing when are creating auto-evals for such models since I didn't actually see a scientific hypothesis being tested here about WHY these models might be doing this; simply describing that the models do poorly or not on some things is not a scientific hypothesis.

---

> > > ### Author Response · Authors · 2024-11-25
> > >
> > > We're glad that our responses were helpful.
> > >
> > > "Flowery language": we see your point, but we think the instruction is still clear that the response is supposed to be shortest change (you might call your best friend first in your example, but the fastest way might still be someone else. This is somewhat subjective, which tends to happen when you move from really strict logical formats to more naturalistic sentences. And you might still disagree with this choice. *But* the nice thing about this framework is that each researcher using it can make these choices themselves. If you want to do an experiment on our framework, you can chose to no use this option; or write your own.
> > >
> > > Statistical tests: I'm not sure what statistical test you're suggesting here for something "looking U-shaped", but we can add this before the final draft if you have suggestions. Personally, we find CI's to be visually somewhat more revealing than p-values since p-values are just a binary number where CI's give you a visual idea of the likely range of values. But this is maybe too in the statistical weeds for our purposes. Again, the point here was to show that our framework can conduct these experiments, not necessarily the results of the experiments themselves. Users of our framework can run whatever statistical tests that suit them.
> > >
> > > We observed a decrease in a majority but yes, we should mention that for some of these it's within the CI. We'll revise this.
> > >
> > > Domjan: again, it's a really old effect. If you have a better source for this, we're happy to cite that instead.
> > >
> > > Closed models: I think that we can still have scientific hypotheses and tests about systems where we do not have all the details about the architecture or training or how it was built. For instance: humans. We sometimes can get some intermediate signals like fMRI or EKG's, but usually in the Psych literature, all we do is study the observed behaviors. And in the ML community, these closed models exist but there is a clear need to try to understand them and test them.
> > > For the question of "why" deep learning does what it does: this has been a really big problem even when we have the model weights available. (Like with humans) we have some techniques in the field of Explainability/Interpretability, but we usually don't get really clear "why" answers and there are many difficulties trying to interpret the output of a several-billion parameter non-linear model.

---

> > > > ### Comment · Reviewer_UYz6 · 2024-11-25
> > > >
> > > > (again, let me preface that I appreciate the response, and that this is all very minor at this point as far as the evaluation goes)
> > > >
> > > > I can imagine different statistical tests depending on the particular claim. Probably the simplest one would be to fit a quadratic and check if the parameter in front of the polynomial 2 is significantly different than 0 in the direction you expect (fitting y=a*x^2 + b*x + c, checking if the confidence interval over the 'a' parameter includes 0 or if it is all positive); another option is -- you previously claimed there was a switch point about 15 entities so one could also fit a piecewise linear model with to-be-discovered switch point and show there is indeed a switch point such that before it the linear model is going down, and after the switch point the linear model is going up.
> > > >
> > > > =====
> > > >
> > > > * I also get that the author's point that they are presenting a framework for running experiments rather than making a point about the results of running their own experiments, but if you are trying to 'sell' a new 'product' (so to speak) you presumably want to show that this new product works and is helpful in producing things -- in case case new insights, I feel like a lot of the discussion here is about people pushing you on the insights of the experiments (whether the stats being used to back them up or their contents) and hearing back "well, run whatever stats and conclude whatever you want". It's a bit like hearing about a new cleaning product, asking the makers if it in fact removes stains because it isn't clear from the picture, and being told "well, check for yourself if it removes stains by whatever checking method you like, we're just making a product here".
> > > >
> > > > I'm exaggerating rhetorically for the sake of the point, and I hope you take this comment as trying to be helpful to you in structuring your response and paper, I feel like some of the pushback you are experiencing and the frustration is due to this cross-wire, but try to understand it from the point of view of something who wants to use your 'product'.
> > > >
> > > > ====
> > > >
> > > >  *"We observed a decrease in a majority but yes, we should mention that for some of these it's within the CI. We'll revise this"  if something is within the CI then it isn't an effect, that's the point of running the stats. "We observed a non-significant decrease in the majority of the cases" ==> means you didn't see a decrease. You could try to make the point that, while every decrease is non-significant on its own, summing over all the cases is significant, but for that you would want a hierarchical statistical model.
> > > >
> > > > ===
> > > >
> > > > Domjan: Ok, one last attempt to get the point across, recognizing that it doesn't actually matter for the paper, results, evaluation, discussion, or review, BUT:
> > > >
> > > > Come on, friends, the fact that it is "a really old effect" isn't an excuse to not cite the original work. And sentences like "or a specific cognitive science hypotheses has been tested on an evaluation created for that purpose such as in (Domjan, 2010)" make it sound like Domjan (2010) did that work, which it really didn't.
> > > >
> > > > Again, imagine if someone had a paper talking about transformer architectures citing "NLP for Undergrads (Smith, 2024)" instead of the original transformer papers, including sentences like "transformer architectures have been tested with various fine-tuning, as in Smith (2024)", and when pushed about it respond "Well these are classic architectures at this point, if you have a better source than this undergrad textbook go ahead"

---

> > > > > ### Author Response · Authors · 2024-11-27
> > > > >
> > > > > This pushback is fair -- we went through some literature on this and it's indeed a bit more complex to provide a single specific citation. We will update the text to reflect this.
> > > > >
> > > > > The long-standing effect is that there is an effect of sequence order on recall behavior in general, and that humans have an clear capacity for sequence learning. For example, lists of unrelated words tend to be recalled in order: for lists shorter than 5, they tend to be recalled in their exact order [Dimperio et al 2005], and for longer lists, items are recalled in an order that is locally ordered but globally disordered, with items nearby in time likely to be recalled together [Sederberg et al., 2010]. A large amount of modeling work has considered how memory for sequences might be implemented [Howard & Kahana 2001, Botvinick & Plaut 2006, Mongillo & Tsodyks 2024].
> > > > >
> > > > > As mentioned, the design of transitive inference experiments is often predicated on the assumption that sequences of out-of-ordered information are harder to reason about [Behrens et al 2018], likely built on humans evident capacity for sequence learning at these capacity levels (above). However, it is not clear this has been directly tested for transitive inference specifically.
> > > > >
> > > > > Steirn et al 1995 looked specifically at presentation order, but in the rather different context of value transfer theory (specifically, can pigeons learn to transfer value transitively given episodes in which A was rewarded over B, then B rewarded over C, etc). Another substantial deviation from our setting is that the pairs are repeated more than once, as it takes >1 trials for the animal to demonstrate knowledge of the relationship. Other work that has been proposed to account for the dynamics of transitive inference is betasort, which hypothesizes that compared entities are dynamically mapped to a ranking that is updated with each encountered pair [Jensen et al 2015]. This algorithm would naturally show a preference for ordered stimuli, as there is no updates needed as information comes in. However, this algorithm is importantly not verified on experiments that vary the presentation order: rather, it captures terminal position effects, symbolic distance effects, and transitive inference better than Q-learning (an RL model with no structure learning, making this comparison less about what kind of structure learning is occuring and more about whether it is occuring at all).
> > > > >
> > > > > We'll modify our presentation of this paper to be considerably more circumspect: in particular, we'll note that there are noted effects on memory order that favor sequential structures [Dimperio et al 2005, Sederberg et al 2010], that there is assumed challenge of reordering stimuli in the design of transitive inference experiments [Steirn et al 1995, Behrens et al 2018], but that the effect of presentation order on this kind of one-shot transitive inference performance has not been (to our knowledge) specifically studied.
> > > > >
> > > > > Krystal Dimperio, Kelly Addis, and Michael Kahana. A comparative analysis of serial and free recall. Memory & Cognition, 33:833–839, 08 2005.
> > > > >
> > > > > Sederberg, P., Miller, J., Howard, M., & Kahana, M. (2010). Temporal contiguity between recalls predicts episodic memory performance. Psychonomic Bulletin and Review, 38, 689–699.
> > > > >
> > > > > Marc W. Howard, Michael J. Kahana. A Distributed Representation of Temporal Context. Journal of Mathematical Psychology, Volume 46, Issue 3, 2002, Pages 269-299,
> > > > >
> > > > > Botvinick MM, Plaut DC. Short-term memory for serial order: a recurrent neural network model. Psychol Rev. 2006 Apr;113(2):201-33. doi: 10.1037/0033-295X.113.2.201. PMID: 16637760.
> > > > >
> > > > > Mongillo, G, Tsodyks M. Synaptic Theory of Working Memory for Serial Order. BiorXiv. 2024
> > > > >
> > > > > Behrens TEJ, Muller TH, Whittington JCR, Mark S, Baram AB, Stachenfeld KL, Kurth-Nelson Z. What Is a Cognitive Map? Organizing Knowledge for Flexible Behavior. Neuron. 2018 Oct 24;100(2):490-509. doi: 10.1016/j.neuron.2018.10.002. PMID: 30359611.
> > > > >
> > > > > Steirn, J. N., Weaver, J. E., & Zentall, T. R. (1995). Transitive inference in pigeons: Simplified procedures and a test of value transfer theory. Animal Learning & Behavior, 23, 76-82.
> > > > >
> > > > > Jensen G, Muñoz F, Alkan Y, Ferrera VP, Terrace HS. Implicit Value Updating Explains Transitive Inference Performance: The Betasort Model. PLoS Comput Biol. 2015

---

> ### Author Response · Authors · 2024-11-22
>
> Use of Closed Models
>
> "publicly available language models" doesn't mean "open"
> “It would be good if the authors tested their frameworks on open models….”
> “difficult to assess what is actually going on”
>
> We should have been more clear on this. All of the models are publicly available to run. We do evaluate against Gemma family models, which are open-source. Additionally, in response to this comment and another reviewer’s concerns, we have also added two Mixtral open-source models to our evaluations to add more models overall and specifically more open models to our experiments.
>
> But I think the larger point is concern about what our analysis can say if we don’t have access to the model, and I do want to push back on that a bit. What I would say to that is that there is in fact a lot that we can learn from the models final outputs and our framework specifically actually can help with this. Because we are not just getting accuracy on a static dataset, we are in fact varying the dataset itself along various dimensions on the x-axis (such as number of entities, or if outputs are in order, or symbolic distance) and looking at the overall trends of those results. I think I would go a bit further and say that even when we have full access to a models’ weights, this does not actually make the models necessarily interpretable as the size and complexity of these models makes them extremely hard to interpret even with the weights.
> I think it’s right to be concerned about what closed-models mean to the scientific community, but I don’t think it’s fair to take that frustration out on papers that are trying to do science despite that by creating evaluations that can tell us things even just from model outputs.
>
> Connections to human reasoning and psychology
>
> General
>
> “The connections to human reasoning and psychology are welcome and overall ok, but there are many places where they could be improved (see more details below). Also, I found several of the cases of arguing along the lines of "the models are doing poorly and people also do poorly so maybe they're the same" to be speculative at best and not digging into the phenomena, again see more details below.”
>
> We respond to the specific phenomena below. In general, we agree (and note in the discussion) that comparisons with humans, while informative, should be interpreted with caution, as there is always ambiguity about whether the origin of similarities/differences are due to data, architecture, optimization, or coincidence. We made an effort to be careful to avoid concluding that models and humans are the same because of similar behavior, and rather describe results in terms of whether behaviors are similar. However, we’ll endeavor to make this more explicit in the paper, and the review process has been helpful in identifying places where we can make this clearer.
>
> We also note that focusing on failure modes in particular has two advantages. First, failure modes are more informative to the ML literature, as they point to where there is a need for new methodological innovation. Second, failure modes are more informative in pointing out biases in a reasoning process.
>
>
> Is Friendship symmetric / non-symmetric relations
>
> We note that this is not unprecedented in the literature on the psychology of relational reasoning when using social networks to probe relational reasoning – for example, Kumaran & Maguire 2005 use the term “friends” to describe individuals joined by a social connection (although this work is on relational memory not social psychology). However, we will be more explicit that we are making the assumption that "friendship" is symmetric, although of course there are asymmetries in real world relationships. If it might be confusing or misleading to readers, we’re also happy to change our terminology to “social connection.”
>
> We point out that navigating an undirected graph is very much a type of relational reasoning – arguably the paradigmatic example – and substantial work in this field looks at exactly these types of problems [Battaglia et al 2018, Whittington et al 2020, Eichenbaum & Cohen 1988]. Spatial relations, asymmetric relations, and typed relations also comprise different types of relations and fall within the field as well, and comprise more complex and elaborate forms of relational reasoning.
>
> It would be interesting and exciting to consider rivalries, hierarchies, family relations, etc. Some of these are already implemented in our framework (but we have not conducted any experiments using these yet) and this would be an interesting area for future research.

---

> ### Author Response · Authors · 2024-11-22
>
> Choice of psych citations
>
> “Given the importance of relational reasoning that the authors emphasize and I agree with, I found it a bit odd that when we do get to references to back it up in humans it is to a 1976 paper on a specific phenomena of symbolic distancing, a 2010 general textbook, and a 2018 neuroscience paper on episodic memory in the hippocampus. This feels like a "grab bag" of disconnected things."
>
> We will update the text to better reflect the more coherent narrative we had in mind. We were generally interested in drawing inspiration from relational memory and reasoning studies, and had a specific focus on relational reasoning studies that challenge memory. Thus, we mimicked psychology experiments that tax capacity of the relational structure or the reasoning problem, create conflict between relational information that is in-weight versus presented in context, and vary the semantic domain (comparisons, social, syllogistic). We also included a few other studies not anchored to existing psychological experiments to highlight the expressiveness of the dataset generating framework, which was the main contribution.
>
> We endeavored to contextualize these studies in the psychological literature described in the background section, and will improve this description and reiterate it throughout the results. The background is where we explained how the experiments we used fit in (and we also include a denser set of citations that more thoroughly illustrates different studies and hypothesized neural mechanisms of relational reasoning in the brain).
>
> Reliance on Domjan
>
> We acknowledge this is unusual, but given the effect of presentation order on transitive inference is a well-established phenomenon, an intro psych textbook is an appropriate citation. Indeed multiple studies address how relational information might be chained or re-ordered in the brain without even referencing a dependence on ordering in the first place [Kumaran 2012, Liu et al 2018, Behrens 2018], and other more recent papers also cite this textbook (eg Hotta et al 2015). We’re happy to include more recent citations (Hotta et al 21015, Steirn et al 1995’s) if recommended. We also note that although it was cited more than once, we relied on it exclusively for presentation order effect.
>
> Stupple & Ball / Evans
>
> We apologize for this; we think that this happened as we were condensing our original related work and grouped in human studies with LLM studies. We’ve fixed this in the draft.
>
> Inconsistent/Consistency detection Experiment
>
> We will work on integrating the experimental sections together better and being more clear which experiments are directly inspired by related cognitive science work and which are more creative demonstrations of the framework (without a specific prior experimental work). Inconsistency detection is an important contribution of our dataset because it demonstrates the flexibility of being able to quickly configure ReCogLab to measure specific abilities in language models.
>
> Non LLM work (Inductive Logic Programming)
> That is a good point, we will add those references.
>
>
> Symbolic distance effect
>
> “The authors should probably face head-on the fact that spatial distancing in these models isn't working like it does in people.”
>
> Yes, we should have been more clear that while there are some commonalities in the u-shape but indeed the effect in humans is as you describe which is not exactly what we observe in our experiments. We have updated this.
>
> Nonsense relations
>
> “'nonsense relations' like 'Benny blexed Kelly' is not 'inconsistent' with the real world in the same way that 'iron is lighter than air'.”
>
> We apologize for the confusing wording, we will clarify what nonsense relationships mean and use a better description in our draft. Congruent statements are statements that agree with real-world physical assumptions about two objects. Incongruent statements are statements that disagree with real-world assumptions. Random-String (the nonsense relations setting) replaces the physical objects with randomly generated string identifiers which disconnects the relationship from any semantically meaningful prior knowledge. In the instructional example we gave, we have two nonsense objects (glarbs and bojaks) with no prior on size and a relationship descriptor (Glarbs are larger than bjoaks). One possible random-string relationship we could’ve generated is “UYz6 is older than P5ze”. Because of anonymity, we have no prior knowledge of these two reviewers and are therefore disconnected from any knowledge bias in how we solve these transitive inference problems.
>
> Other issues/questions
>
> Citations in Intro
>
> We bring this related work later in background, but you’re right, we should have cited them when we refer to this in the intro as well. We have added those citations as well as for the proposition For “[w]hen humans remember events….” yes, we have also updated the paper to include references supporting this.

---

> ### Author Response · Authors · 2024-11-22
>
> Grounding
>
> “[I]t wasn't entirely clear to me how you check for grounding in a generalizable way.”
>
> One could always crowdsource the labels. In general inspection we find that they're typically consistent with our personal measure of objects. It also fulfills another niche because it specifically presents examples that are incongruent with the language models knowledge (and not just incongruent with reality which would be a different but related thing to measure)
>
> Capacity Ranges
>
> A: For JSON family specifically we did not go beyond this range due to cost (this dataset specifically has much larger numbers of tokens and it would have blown through our budget to go to 100 as in the others). For the other experiments, the main goal was to cover a range of values.
>
> Other writing / typos
>
> Thank you for these, we will fix them in the draft.
>
> Prompting details
>
> See the updated Appendix B for more details and example prompts.

---

### Official Review · Reviewer_P5ze · 2024-11-02

**Soundness:** 3
**Presentation:** 3
**Contribution:** 1
**Rating:** 5
**Confidence:** 4

**Summary:**

This article introduces ReCogLab, a flexibly generated dataset for quantitatively measuring relational memory and reasoning performance in LLMs, and conducts several experiments using the framework. ReCogLab offers a fully customizable and automatically generated dataset, enabling researchers to delve into the nuances of cognitive processes.

The paper begins by introducing ReCogLab. Examples generated by ReCogLab come from one of four tasks: Comparison, Social Networks, Syllogisms and JSON Families. Each example includes context C, question Q and answer(answers) A. Every aspect of the data generated can be controlled via a configuration file specifying parameters, which allows for the creation of datasets with varying length and complexity. And for each specific cognitive probe, ReCogLab generates 50 validation examples to select the best prompt and 1000 test examples for comprehensive evalution. Notably, the authors decoupled the prompt and parsing hyperparameters from the LM to minimize the impact of prompting on LMs’ performance.

Then, the authors conduct several experiments to benchmark relational reasoning performance across different models and problem complexities and explore how reasoning performance depends on certain features. The article looks at transitive inference, congruency, the symbolic distance effect, identifying logical inconsistencies and indeterminate reasoning. These experiments were all made with little additional effort using the ReCogLab framework by altering the configuration files. The findings reveal that many human cognitive phenomena are mirrored in LLMs.

**Experiment 1: Transitive Inference**

Purpose:

To evaluate models’ ability to reason accurately about associations presented in context and measure whether there was a dependence on presentation order, and further measure how the ability depends on complexity measures of the reasoning problem.

Results:

1.(Comparison and Syllogisms) Randomizing the order causes noticeable degradation while reversing the presentation order affects performance in many but not all cases.

2.(Syllogisms)The authors speculate that consistent presentation order may bias the model toward approximate solutions.

3.Performance generally degrades as the complexity of the problem increases across all models.

**Experiment 2: Congruent and Incongruent**

Background:

Logical reasoning is more accurate when logical premises are congruent with real-world facts for LLMs(as it is for humans).

Purpose:

To investigate the impact of congruence on language model performance.

Results:

Using congruent statements outperforms similar comparison problems constructed with incongruent statements.

**Experiment 3: Symbolic Distance Effect**

Purpose:

To validate that LLMs will also show a symbolic distance effect.

Results:

For models that perform above chance, there is a clear positive symbolic distance effect that starts when the symbolic distance is greater than 15.

**Experiment 4: Identifying Inconsistent Premises**

Purpose:

To explore whether LLMs have the ability of identifying when a premises is logically inconsistent.

Results:

Larger LMs perform better on detecting inconsistent statements.

**Experiment 5: Indeterminate Reasoning**

Purpose:

To evaluate models’ ability to understand when there’s insuffient information to draw a conclusion.

Procedure:

The authors start with comparison problems which contain a fixed label set of Yes or No. Then they modify the comparison problems from a linear chain to a random tree generation while still asking questions about two random nodes, which provides insufficient context.

Results:

There exists a bias toward incorrectly answering \textquotedblleft yes" on logical prompts when it is uncertain and reporting uncertainty when in fact it is unknown.

**Strengths:**

**Flexibility and universality**

ReCogLab has significant advantages over the previous dataset in terms of flexibility and universality. Such a dataset would bring many conveniences to research in cognitive sicence and holds significant practical value.

**Good breadth**

This article demonstrates a good breadth, reprising experiments from multiple aspects of cognitive science, and measuring the ability of multiple LLMs across various dimensions.

**Weaknesses:**

**Lack of evaluation of dataset quality**

When evaluating the quality of a dataset, people commonly employ methods such as statistical analysis, visualization, cross-validation, and expert review. However, in this article, we only see qualitative expressions about the performance of ReCogLab, such as its comprehensiveness and flexibility. And I’m wondering would it be possible to add some quantitative indicators to more strongly demonstrate the quality of ReCogLab or compare ReCogLab with those originally used in the same or similar experiments to illustrate its advantages?

**Lack of detailed experimental procedures**

The description of the experimental procedures could be more detailed. It did not provide a detailed explanation of how ReCogLab was used in these experiments and how exactly the parameters were set.

**Unclear motivation**

The authors haven’t provided evidence for their classification (comparison, social networks, syllogism, json families). What’s more, the decision to test all tasks as a whole is questionable, especially considering that previous research has addressed these aspects individually. The necessity of compiling these tasks for collective testing is not immediately clear, and this approach may need further justification.

**Lack of insights**
The authors simply test LLM without delving into a more detailed analysis of each aspect, which results in superficial analyses. The authors should perform deeper analysis, such as for the transitive inference, the authors should offer an explanation of why the change of order matters for LLM, for congruent why is it important (whether this phenomenon comes from data or from training algorithm or transformers architecture), ...

**Lack of novelty**
1. In Experiment 1, the authors' assertion that "premise order matters" was previously identified in the article "Premise Order Matters in Reasoning with Large Language Models" published on May 28th. In Experiment 2, the earlier paper "Language Models, like humans, show content effects on reasoning tasks" demonstrated the importance of congruence in reasoning. Furthermore, there appears to be no significant distinction between relational reasoning and general reasoning. Could the authors compare with those above-mentioned papers to address the novelty of this paper? Furthermore, could the authors talks about the difference between relational reasoning and general reasoning?

 2. Additionally, in Experiment 4, the detection of inconsistency is an old topic, and it comes as no surprise that Large Language Models (LLMs) are capable of performing this task, could the authors demonstrate some new discoveries in their experiments? Moreover, this topic bears a striking resemblance to Knowledge Base Question Answering (KBQA), can the authors provide the difference between the topic of relational reasoning and KBQA?

**Some editorial errors and missing citations**

Introduction paragraph 1: While recent work...

Citations are missing.



Introduction paragraph 2: From this literature...

The authors didn’t point out what “this literature” refers to.

Introduction paragraph 3: In this work, we aim to provide an evaluation framework that allows for the systematic evaluation or LLMs on relational memory and investigation of different possible effects(many inspired from the cognitive science literature).

Maybe it means “of”?

Background paragraph 1: $A > B$, $B > C$, $B > C$, $C > D$, $D > E$

Perhaps “$B > C$” is repeated?

**Questions:**

As seen above.

---

> ### Author Response · Authors · 2024-11-22
> **Response**
>
> Thank you for your review! We hope that you take a look at our general response and we reiterate that our main goal is to create a framework which can generate datasets for evaluating language models along many different dimensions. We are glad that you can see some of the practical value our proposed framework has to offer,
>
> We will now go through your questions and other feedback, covering in particular your main points for improvement: framework details, why these tasks, improving connections to human reasoning and psychology and statistical claims. Please follow up with us for any additional points of clarification.
>
> Novelty of dataset/experiments
>
> We address this point in our general response. In particular, our novelty is in the flexible, generative framework for datasets. Similarities in several of our experiments are deliberate to show that our framework is flexible to generate datasets to study a variety of cognitive-inspired (or non-cognitive inspired) effects, both known and new. For the particular two experiments mentioned: in the general response we go over how our experiment differs from "Premise Order Matters in Reasoning with Large Language Models” and again, we do not claim that this was an undiscovered effect. For the consistency experiment
>
> Unclear Motivation
>
> We appreciate the feedback on our motivating reasons. Our goal isn’t to develop a static benchmark for researchers to evaluate and hill-climb on (there’s plenty of those being submitted every year now). We are proposing a new way to reinvent the evaluation process. Instead of using static test examples that are hand collected to test for specific patterns of cognition in a language model, we are designing a framework that can be easily customized to test various hypotheses about language models. This allows us to measure the effect of different variables easily. We also note that this pattern of evaluation lends itself to other kinds of scientific safeguards like overfitting, training data contamination, and step-function-esque emergent behaviors through scaling difficulties. See our general response for more details.
>
> Lack of evaluation of dataset quality/Why put all these tasks together, why not just concatenate previous datasets
>
> As mentioned in the general response, ReCogLab isn’t a static dataset, but a generative framework for quickly prototyping a dataset to test hypotheses. We agree that there’s lots of really interesting downstream analysis for both understanding the cognitive biases and underlying mechanisms of a language model. Previously, these published works in both cognitive science and machine learning have focused on verifying a single, specific hypothesis by painstakingly assembling test examples to measure the effect size. Our single sandboxed framework allows us to discover similar findings as previous works (such as the premise ordering) while also exploring other cognitive hypotheses like Acquiescence bias. Also the paradigm of a generative framework instead of a static dataset decreases the chance of overfitting and LLM training contamination as the exact example and answer is not easily scrapable without intentionally running the framework. This is a key distinction and why we don’t simply concatenate hand-curated datasets.
>
> Lack of detailed experimental procedures
>
> We have added additional information to the appendix on how specific configurations of the task we experimented on were generated in Appendix A.  We also added Appendix B which includes a discussion on the prompt validation procedure used to debias prompt overfitting and the list of prompts we validated on. We hope that this clears up any questions about how the experimental results were collected.
>
> Tie-in to KBQA
>
> We have added references to KBQA problems.There is indeed a strong connection through multi-hop relational reasoning. KBQA would be an excellent candidate to incorporate into ReCogLab task too as the structure of the problem matches all the criteria that we picked for the first four tasks (easy to generate random examples from a fixed domain of facts/priors, easy to verify the answers, unambiguous rules of logic to solve the problem).
>
> Why (these specific) tasks
>
> Please see our general response where we try to answer this question. We have also added some additional text in the draft to make this clear to future readers as well.

---

> ### Author Response · Authors · 2024-11-22
>
> Lack of Insight
>
> We agree that detailed analysis on mechanistic interpretation of language models is important too. Exploring whether these cognitive phenomena are caused by data, training algorithms, or transformer architecture is a topic deserving of its own independent paper built on top of ReCogLab. Our work here focused primarily on how ReCogLab could empower researchers to explore these mechanistic interpretations with less activation energy. As an example, prior works like “Premise Order Matters in Reasoning with Large Language Models” focused on a single cognitive effect, but a sizable part of their contribution came in constructing R-GSM from GSM-8K (notably still a static benchmark and high risk of contaminating a language model’s training data).
>
> More detailed analysis
>
> Yes, we can expand more in the draft. We do want to reiterate that our motivation is creating a framework that lets you analyze a wide variety of problems rather than specifically doing these analyses. But since we have done the experiments to make that point, we can also add more details.
>
> Furthermore, could the authors talks about the difference between relational reasoning and general reasoning
> A lot of attention has been put on “general reasoning” as a method for extracting higher-order capabilities. Our reasoning problems focus specifically on combining information about how two entities are related to answer higher questions. Many general reasoning problems can be formulated as relational reasoning or contain a relational reasoning aspect, but there are some reasoning primitives that don’t match these patterns. For instance our information context contains perfect information to answer each question. On the other hand, some reasoning problems require implicit knowledge. The question “It’s raining, should I bring an umbrella” has a natural conclusion but requires a lot of unspoken, implied assumptions like 1. It’s raining where I am located, 2. an Umbrella is an effective device for staying dry, 3. I will spend lots of time outdoors.
>
>
> Citations in Intro
>
> For the first citation issue: “while recent work”, we bring up the related work later in background, but you’re right, we should have cited them when we refer to this in the intro as well. We have added those citations. For the second one “From this literature” we are a bit confused by your comment; we cite (Domjan, 2010; Moyer & Bayer, 1976; Koster et al., 2018) in this same sentence. If you could clarify your confusion we can try to address your concern.
>
> Grammar/Typos
>
> Thank you for pointing these out, we have fixed these in the paper.

---

### Author Response · Authors · 2024-11-22
**General Response**

First, we would like to thank all of the reviewers for their time and helpful comments. They were genuinely really helpful for us in adding references, experiments and clarifications. We have incorporated this feedback into our draft and believe that the paper is better as a result.

We are glad that many of the reviewers found significant value to our framework.
P5ze says “ReCogLab has significant advantages over the previous dataset in terms of flexibility and universality. Such a dataset would bring many conveniences to research in cognitive science and holds significant practical value.” Similarly, P3cs says our dataset generation approach is “exactly where evaluation should be going in AI and machine learning.” G6qF recognizes the importance of benchmarking aspects of cognition “Examining the capacities of LLMs is important, and it is important to understand the more difficult aspects of potential cognition such as relational reasoning. Establishing benchmarks is an important part of building this research program.”

Several reviewers are also happy with the breadth of the experiments enabled with our framework and our careful analyses.
P3cs: “Finally, the analyses are comprehensive and balanced. The authors are cautious not to prematurely ascribe human-like cognitive capabilities to these systems, noting that there is more work to be done to show this. This is a refreshing contrast to much of the overzealous AI hype from other (non-cognitive-science) researchers in the field.”
P5ze: This article demonstrates a good breadth, reprising experiments from multiple aspects of cognitive science, and measuring the ability of multiple LLMs across various dimensions.

Next, we would like to respond to some common points of misunderstanding and more clearly restate our motivation and goal for this work. We think that reviewer P3cs stated it quite nicely:
“[T]he production of a sandbox testbed, rather than a static, fixed benchmark, is exactly where evaluation should be going in AI and machine learning. ReCogLab is a toolbox for creating nuanced cognitive experiments, rather than a dataset that someone can take off the shelf and apply without much thought.”
This was exactly our motivation! The goal of ReCogLab was not to introduce a single new, static dataset for evaluating LLMs. The goal was to create a framework which can generate datasets for comparing and evaluating language models along many different dimensions. Imagine that you are a Cognitive Scientist who has a hypothesis about how LLMs work motivated by work in relational reasoning in humans. Or, imagine that you are an LLM user or designer and you want to know how well different LLMs respond in different scenarios. What you want to do is isolate the variable you care about, such as the complexity of the underlying graph or the order in which statements appear. What prior work has done in the past is create single, one-off datasets, often hand-written, to study each particular effect and then validate results on a number of different models. What we are instead trying to do is create a framework where this researcher can instead use our framework and quickly set up this particular experiment and get results fast.

With this motivation in mind, we hope this clarifies a few points for reviewers. P5ze asks what is “the necessity of compiling these tasks for collective testing” and similarly G6qF asks “The authors main claim is that there isn't a set of all these benchmarks together, but one should simply stitch together other benchmarks as a set.” We think this misunderstands what it is that we are doing. Our paper is not a static collection of datasets; it is a framework for generating new datasets. We suspect that the fact that we deliberately generated some datasets and experiments to be similar to prior work’s experiments may have led reviewers to think this, but our motivation here was to show that our framework is able to generate datasets that include all of these prior experiments but also that our framework is able to generate new datasets which prior works did not and could not. So to answer the question “why not just staple together previous datasets” the answer is that if you want to: study the same parameter of interest (e.g. ordering) but also want to simultaneously vary another (e.g. context size) or study that parameter with much longer context windows, the prior datasets made for studying ordering will not work! Or, if there is a parameter you want to vary, a researcher can quickly add a new condition to the generation in a few lines of code and re-generate new datasets studying that new parameter. These things are simply not possible by using prior static datasets. The goal here is to minimize the time the researcher needs to set up the datasets for an experiment and make extremely specific specifications of the data possible, which we believe our framework provides.

(continued)

---

> ### Author Response · Authors · 2024-11-22
> **General Response (continued)**
>
> Reviewers UYz6 and P5ze both also brought up questions about the novelty of our experiments and where they fit into the literature and we hope this clearer statement of our motivation helps explain this. In both cases, the reviewers point out several of our experiments have direct parallels (or similar motivations) to prior works. And, again, this was deliberate. Our goal was to show the flexibility of our framework to incorporate many of these prior experiments as well as to vary them and build new experiments. We also ran new variations of experiments on different tasks. For instance, as P5ze points out we did also study premise order as did a recent work "Premise Order Matters in Reasoning with Large Language Models,” but in our case we ran it on different tasks from that paper (they studied on word problems), which, if we want to know if premise order is a general trend is very useful. But unlike in that work, if we wanted to study the effect of premise order for very long context examples, you could not do that in their dataset because it is fixed to the problems in GSM8K, but in our framework, you could generate whatever number of statements you wanted. We hope this clarifies this point.
>
> Another question that several reviewers had was why/how we chose our specific tasks for our framework? Why these four tasks in particular (comparison, social networks, syllogism, JSON families) and not others? (Also for clarity please note that task != dataset. A dataset is a static collection of examples {X_1, Y_1, …. X_N, Y_N} while a task in our framework is a category of problem which with different parameter values can generate any number of datasets).
> First, some requirements for all of these tasks were that they are able to be generated statically and have easy to verify answers (so you need to be able to generate them automatically with code using templates). Next, as UYz6 suggests, our goal is to cover many different kinds of relational reasoning. Many of these aspects of relational reasoning are well studied in humans such as social networks and syllogisms. Similarly, comparisons (and in particular effects such as the symbolic distance effect) made comparison a type of task that was natural to add. We also want to highlight P5ze’s comment on Knowledge Based Question and Answering which we believe would make an excellent new domain to ReCogLab. Beyond these, we also wanted to allow and show how users of the framework could be creatively expanded by, for instance, seeing how if the data is stored in a formatted representation, how does this affect things. We introduce and show experiments on these tasks, but the goal is for users of our framework to add their own tasks and we hope they do! We hope this clarifies, and we agree with UYz6 that we should have made this more explicit. We have added a more explicit motivation for these tasks in the draft.
>
> We thank reviewers again for their time and feedback and will respond to their specific comments in the individual responses.

---

### Author Response · Authors · 2024-12-02

Today is the last day reviewers can respond. We ask that you kindly acknowledge our responses if you haven’t already and ask any additional questions you need to finalize your review.

---

### Meta-Review · Area_Chair_ovew · 2024-12-20

**Metareview:**

The paper presents a framework, called ReCogLab, for generating synthetic evaluation data that permits careful probing of the various aspects of relational reasoning (in the psychological sense). RecogLab enables flexible testing, revealing that many human-like cognitive effects are also present in LLMs, offering a valuable tool for advancing AI research on human-like reasoning capabilities.  Overall, the reviewers agree that testing LLMs on their basic understanding of relations is useful for our understanding of both AI models and human cognition. However, the reviews also raised downsides. As one reviewer points out, the authors present the generative framework itself as a major contribution without providing much detail on it. While the authors provide some details in the appendix, they are not very detailed. Consider the rejection sampling. How exactly is this done? What does the pseudo-code look like? Are there important hyperparameters?Additionally, one may provide an (anonymous) code repository already for the reviewing process. Regarding details on the dataset, however, the appendix looks very nice. Thanks. Moreover, a discussion with one reviewer actually shows how deep ReCogLab can be anchored in the cognitive literature. This should definitely go into the final version so that even the computer scientist can understand the value of ReCogLab. This is even more important as the most critical review makes the point that the design choices of ReCogLab are not clear. For instance, do humans perform differently on these tasks, and might that be a reason that the different benchmarks are needed? However, due to the discussion, e.g. about the literature, this point seems to be somewhat addressed, and hence the paper should rather be accepted.

**Additional Comments On Reviewer Discussion:**

This was for me a very interesting discussion, partly because I was observing a different culture (and view) from a different discipline, namely psychology/cognitive science.   At least I was learning a lot. Topics touched were closed models, design choices, task selection, AI versus cognitive science, etc. Now, indeed, we potentially overrule on negative review, but here the discussion also shows that it helped clarify major issues raised. So overall, I tend to accept the paper.

---

### Decision · Program_Chairs · 2025-01-22

Accept (Poster)